# DET1-mediated degradation of a SAGA-like deubiquitination module controls H2Bub homeostasis

Amr Nassrallah[1†‡], Martin Rougée[2,3†], Clara Bourbousse[2,3], Stephanie Drevensek[2§], Sandra Fonseca[1], Elisa Iniesto[1#], Ouardia Ait-Mohamed[2], Anne-Flore Deton-Cabanillas[2], Gerald Zabulon[2], Ikhlak Ahmed[2¶], David Stroebel[2], Vanessa Masson[4], Berangere Lombard[4], Dominique Eeckhout[5,6], Kris Gevaert[7,8], Damarys Loew[4], Auguste Genovesio[2], Cecile Breyton[9], Geert De Jaeger[5,6], Chris Bowler[2]*, Vicente Rubio[1]*, Fredy Barneche[2]*

[1]Centro Nacional de Biotecnología, CNB-CSIC, Madrid, Spain; [2]Institut de biologie de l'Ecole normale supérieure (IBENS), Ecole normale supérieure, CNRS, INSERM, Université PSL, Paris, France; [3]Université Paris-Sud, Orsay, France; [4]Centre de Recherche, Laboratoire de Spectrométrie de Masse Protéomique, Institut Curie PSL Research University, 75005 Paris, France; [5]Department of Plant Systems Biology, Ghent University, Ghent, Belgium; [6]VIB Center for Plant Systems Biology, Ghent, Belgium; [7]Department of Biochemistry, Ghent University, Ghent, Belgium; [8]VIB Center for Medical Biotechnology, Ghent, Belgium; [9]Université Grenoble Alpes, Institut de Biologie Structurale, Grenoble, France

**\*For correspondence:**
cbowler@biologie.ens.fr (CB);
vrubio@cnb.csic.es (VR);
barneche@biologie.ens.fr (FB)

[†]These authors contributed equally to this work

**Present address:** [‡]Biochemistry Department, Faculty of Agriculture, Cairo University, Giza, Egypt; [§]Institut of Plant Sciences Paris-Saclay (IPS2), UMR 9213/UMR1403, CNRS, INRA, Université Paris-Sud, Université d'Evry, Université Paris-Diderot, Paris, France; [#]Salk Institute for Biological Studies, La Jolla, United States; [¶]Weill Cornell Medicine - Qatar (WCM-Q) Education City, Doha, Qatar

**Competing interests:** The authors declare that no competing interests exist.

**Abstract** DE-ETIOLATED 1 (DET1) is an evolutionarily conserved component of the ubiquitination machinery that mediates the destabilization of key regulators of cell differentiation and proliferation in multicellular organisms. In this study, we provide evidence from Arabidopsis that DET1 is essential for the regulation of histone H2B monoubiquitination (H2Bub) over most genes by controlling the stability of a deubiquitination module (DUBm). In contrast with yeast and metazoan DUB modules that are associated with the large SAGA complex, the Arabidopsis DUBm only comprises three proteins (hereafter named SGF11, ENY2 and UBP22) and appears to act independently as a major H2Bub deubiquitinase activity. Our study further unveils that DET1-DDB1-Associated-1 (DDA1) protein interacts with SGF11 *in vivo*, linking the DET1 complex to light-dependent ubiquitin-mediated proteolytic degradation of the DUBm. Collectively, these findings uncover a signaling path controlling DUBm availability, potentially adjusting H2Bub turnover capacity to the cell transcriptional status.
DOI: https://doi.org/10.7554/eLife.37892.001

## Introduction

DE-ETIOLATED 1 (DET1) is an evolutionarily conserved factor in multicellular organisms, acting in most instances to regulate gene expression through ubiquitin-mediated protein degradation. *DET1* null mutations are lethal in plants (*Miséra et al., 1994*; *Pepper et al., 1994*), Drosophila (*Berloco et al., 2001*) and Human (*Wertz et al., 2004*). However, viable Arabidopsis knockdown alleles identified in genetic screens for mutant plants displaying a constitutive photomorphogenic phenotype (i.e. de-etiolated) have unveiled that DET1 is a central integrator of light signaling in plants (*Chory et al., 1989*; *Pepper et al., 1994*). The Arabidopsis *det1-1* mutation affects the expression of thousands of nuclear genes (*Ma et al., 2003*; *Schroeder et al., 2002*), partly because

proteolytic degradation of the proto-photomorphogenic transcription factor HY5 is abolished in this background, thereby mimicking the presence of light on the transcriptional program (*Osterlund et al., 2000*). In humans, DET1 also controls the stability of cell proliferation factors such as the Cdt1 DNA replication-licensing factor (*Pick et al., 2007*) and the proto-oncogenic transcription factor c-Jun (*Wertz et al., 2004*). Accordingly, a currently accepted model in both plants and animals is that DET1 is an atypical DAMAGED DNA BINDING PROTEIN 1 (DDB1)-CULLIN4 (CUL4) Associated Factor (DCAF) acting with the small DDA1 (DET1-DDB1-Associated 1) protein to provide specificity to one or more E3 CUL4-RING ubiquitin ligases (CRL4) (*Chory, 2010*; *Lau and Deng, 2012*). For this activity, DET1 and DDA1, together with DDB1 and CONSTITUTIVE PHOTOMORPHOGENIC 10 (COP10) proteins, constitute a substrate adaptor module (COP10-DET1-DDB1-DDA1; hereafter termed C3D) within CRL4 complexes (*Irigoyen et al., 2014*; *Pick et al., 2007*). C3D binding to the CUL4 scaffolding protein is mediated by the core adaptor subunit DDB1 whereas the E2 ubiquitin conjugase variant COP10 likely acts to increase the activity of CRL4 complexes towards specific protein targets (*Lau and Deng, 2012*).

Photomorphogenesis is a developmental switch that initiates upon the first perception of light by young plants reaching the soil surface. This transition triggers the launching of organ growth and the establishment of photosynthesis, most notably through the differentiation of primary leaf (cotyledon) cells (reviewed in [*Casal, 2013*; *Seluzicki et al., 2017*; *Wu, 2014*]). The process involves changes at transcriptomic, epigenomic and nuclear architecture levels (*Bourbousse et al., 2015*; *Charron et al., 2009*; *Sullivan et al., 2014*). While several chromatin modifiers are known to influence light-responsive gene expression, the first direct link between light signaling and chromatin came from the discovery that DET1 has high affinity for nucleosomal histone H2B in vitro and in vivo (*Benvenuto et al., 2002*). By potentially being at the interface between light signaling and epigenome dynamics, a chromatin-level function of DET1 in photomorphogenesis has long been proposed (*Chory, 2010*; *Lau and Deng, 2012*), yet the chain of events leading from light sensing to epigenomic reprogramming is still largely ignored (reviewed in *Barneche et al., 2014*; *Perrella and Kaiserli, 2016*). Here, we unveil a direct link between DET1 and the histone H2B monoubiquitination pathway. This histone post-translational modification (PTM) acts cooperatively with RNA Polymerase II (RNA Pol II) and other PTMs of histones to promote gene expression and the regulation of cellular homeostasis in various organisms (*Shema et al., 2008*; *Smith and Shilatifard, 2013*; *Weake and Workman, 2012*).

Transcription-coupled chromatin remodeling and modification play a fundamental role in the fine-tuning of genome expression. In the sequence of events established in the yeast *Saccharomyces cerevisiae*, the Polymerase-associated factor 1 (PAF1) complex interacts with Ser-5 phosphorylated RNA Pol II and serves as a platform for H2B monoubiquitination by the Rad6 E2 ubiquitin conjugase and the Bre1 E3 ligase. In turn, recruitment of the SAGA (Spt-Ada-Gcn5-acetyltransferase) complex at the Pre-Initiation Complex (PIC) triggers H2Bub de-ubiquitination and phosphorylation of Pol II CTD on Ser-2, mediating the transition towards transcription elongation (*Wyce et al., 2007*). Sliding from the promoter, SAGA is thought to travel along the gene body with RNA Pol II to allow for repeated cycles of histone H2B ubiquitination/de-ubiquitination by opposing Bre1 activity (*Henry et al., 2003*). This is thought to facilitate the progression of RNA Pol II through nucleosomal barriers by influencing DNA accessibility, recruitment of the histone chaperone FACT (FAcilitates Chromatin Transcription) and promoting nucleosome reassembly (*Belotserkovskaya et al., 2003*; *Fierz et al., 2011*; *Fleming et al., 2008*; *Pavri et al., 2006*; *Xin et al., 2009*). Accordingly, SAGA displays a general co-activator activity that promotes transcription at a post-initiation step on most expressed genes in budding yeast (*Baptista et al., 2017*; *Bonnet et al., 2014*), with a pronounced effect on so-called SAGA-bound 'regulatable' genes (*de Jonge et al., 2017*).

In budding yeast and metazoans, SAGA is a ~ 1.8 megaDa gigantic complex made of functionally distinct components comprising a TBP-associated factor (TAF) architectural core module, a TATA-Binding Protein (TBP) module, a histone acetyltransferase (HAT) module, and a H2Bub deubiquitination module (DUBm) (*Grant et al., 1997*). DUBm is an independent structure that functionally requires the Ubiquitin protease 8 (Ubp8) catalytic subunit, the Sgf11 (SAGA-associated factor 11) nucleosome-binding subunit and the small protein Sus1 (Sl gene upstream of ySa1), which binds ubiquitin (*Rodríguez-Navarro, 2009*), while the fourth subunit Sgf73 makes the bridge with core SAGA modules (*Köhler et al., 2008*; *Lee et al., 2009*).

Despite the functional conservation of the SAGA central core subunit ADA2b and the GCN5 histone acetyltransferase (*Benhamed et al., 2006*; *Vlachonasios, 2003*), a structural and functional characterization of plant SAGA complexes has long been missing. Orthologous proteins of Ubp8, Sgf11 and Sus1 or their human counterparts USP22, ATXN7L3 and ENY2 have been identified from the genome sequence of *Arabidopsis thaliana* but Sgf73 appears to be absent (*Moraga and Aquea, 2015*; *Srivastava et al., 2015*). Arabidopsis UBIQUITIN PROTEASE 26 (UBP26) deubiquitinates H2Bub in vivo, impacting expression of the *FLC* developmental regulator (*Schmitz et al., 2009*), but it appears to act independently from SAGA and rather displays a repressive activity on transposable elements (*Sridhar et al., 2007*) and imprinted genes (*Luo et al., 2008*). More generally, homolog proteins of SAGA, Bre1 and PAF1c subunits are known to influence plant development and responses to biotic and abiotic stress, (*Grasser and Grasser, 2018*; *Moraga and Aquea, 2015*; *Van Lijsebettens and Grasser, 2014*), but the structure and function of transcription co-activators, their influence on gene activity and on how they are themselves regulated during cell specialization and adaptation largely remain to be characterized in plant systems. Recent purification of elongating RNA Pol II has indeed shown its tight association with chromatin machineries such as PAF1, FACT and HUB complex subunits in Arabidopsis (*Antosz et al., 2017*). HUB is the only known histone H2B E3 ubiquitin ligase in Arabidopsis, acting as a heterotetramer of the HISTONE UBIQUITINATION 1 (HUB1) and HUB2 proteins (reviewed in *Feng and Shen, 2014*). Although null mutations in *HUB1* or *HUB2* genes abolish H2Bub deposition, knockout plants are viable with only mild phenotypic defects in seed dormancy, cell cycle progression, circadian clock and flowering time control (*Cao et al., 2008*; *Fleury et al., 2007*; *Liu et al., 2007*). Still, abrogating *HUB1* function impairs rapid modulation of RNA levels of light-regulated genes during seedling photomorphogenesis, suggesting that monoubiquitinated H2B is required to attain high expression levels during Arabidopsis photomorphogenesis (*Bourbousse et al., 2012*).

In this study, we first provide evidence that DET1 is essential for the regulation of H2Bub levels over most genes by controlling the light-dependent degradation of a plant H2Bub deubiquitination module (DUBm). This DUBm comprises SGF11, UBP22 and ENY2, the Arabidopsis orthologous proteins of yeast Sgf11, Ubp8 and Sus1, respectively, which, in the absence of a predictable plant Sgf73-like subunit, appears to act independently from SAGA. Although the Arabidopsis genome encodes 14 families of ubiquitin-proteases, altogether representing 27 members (*March and Farrona, 2017*), *UBP22* loss-of-function drastically impairs H2Bub removal, indicating its major H2Bub deubiquitinase activity in Arabidopsis. Our study further reveals that Arabidopsis SGF11 physically links DUBm to the DDA1 C3D subunit in vivo and is subject to DET1-dependent ubiquitin-mediated degradation. Collectively, this study uncovers a signaling path controlling global H2Bub levels, potentially fine-tuning gene transcriptional capacity or 'regulatability' during developmental responses to external cues.

## Results

### *DET1* is required for H2Bub enrichment over most genes

To investigate how DET1 impacts on chromatin status, we tested whether histone post-translational modifications (PTMs) were affected in *det1-1* mutant plants by conducting label-free quantitative mass spectrometry analysis of purified histones (*Figure 1—figure supplement 1*). We identified 15 different peptide sequences bearing one or more differentially modified residues in *det1-1* seedlings, half of them matching histone H2B (*Figure 1—source data 1*). Histone H2B was found to be more frequently acetylated and twice less monoubiquitinated at a conserved lysine residue in the carboxy-terminal tail of 9 different H2B isoforms (*Figure 1A–B* and *Figure 1—figure supplement 1B*). Low H2Bub levels in these mutant plants were reproducibly visualized by immunoblot detection using an antibody that recognizes H2B histone isoforms and their slower migrating monoubiquitinated forms in chromatin-enriched samples (*Figure 1C and D*). This defect was also confirmed by immunoprecipitation of MYC-tagged ubiquitin proteins from wild-type and *det1-1* plants followed by detection of histone H2B proteins (*Figure 1—figure supplement 1C*). Finally, introgression of the *hub1-3* mutation in the *det1-1* genetic background indicated that remaining H2Bub marks in *det1-1* plants are deposited via the HUB pathway (*Figure 1D*). The mass spectrometry analysis also indicated an increase of histone H3 methylation on Lys-27 or Lys-36, but no robust differences in

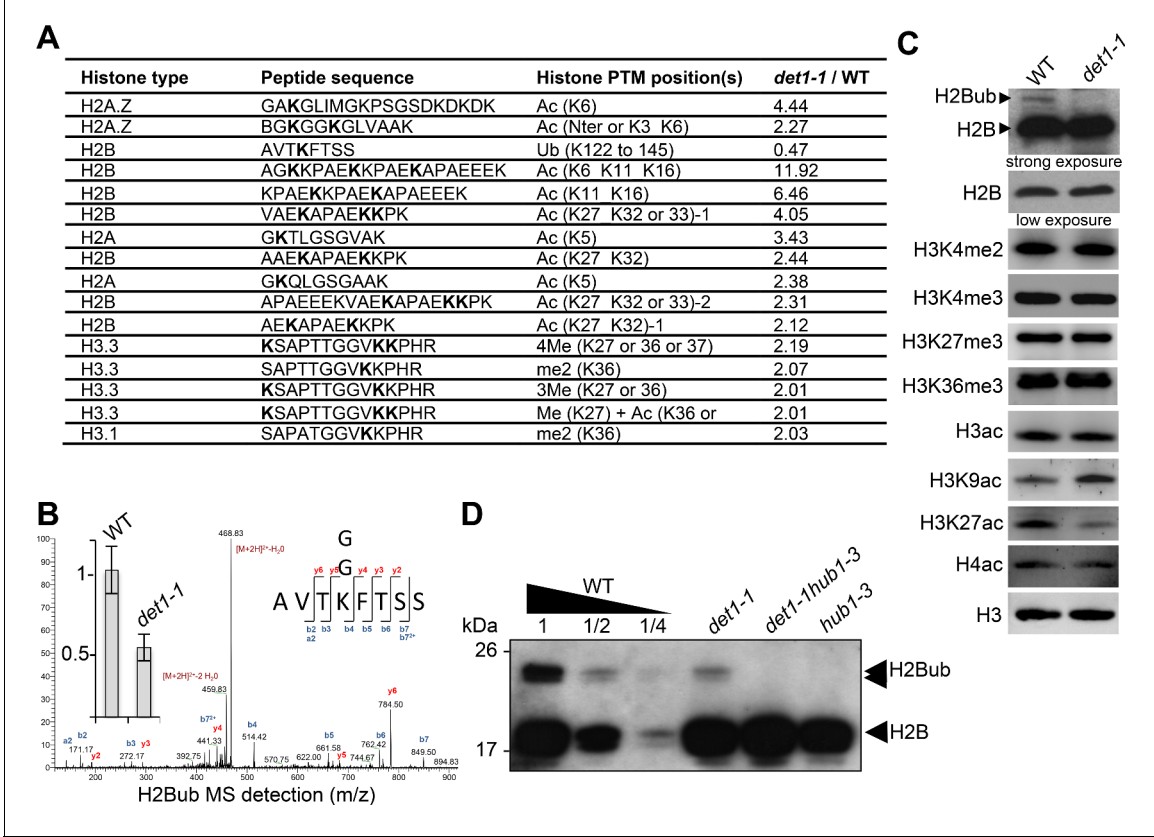

**Figure 1.** *DET1* loss-of-function triggers a massive reduction of H2Bub. (**A**) Mass spectrometry detection of differentially modified histones using non-modified histone peptides as internal references. Differentially modified residues are shown in bold. Several candidate residues are bolded when the precise position of the differentially modified residue could not be determined. Only significantly modified peptides from 3 biological replicates are shown (p-Val <0.001;>2 fold change). Histone isoforms were annotated according to *Talbert et al. (2012)*. Ac, acetylation; Ub, ubiquitination; Me, methylation; xMe: x methylations on a unique or distinct residues; me1, monomethylation; me2, dimethylation; me3, trimethylation. (**B**) Representative MS/MS spectra of histone H2B monoubiquitination at a the conserved lysine residue in the AVTKubFTSS peptide known to be monoubiquitinated in plants (*Bergmüller et al., 2007*; *Sridhar et al., 2007*). The fragmentation spectrum is derived from a trypsin-digested position matching the HTB1/2/3/4/6/7/9/10/11 isoforms. The ubiquitinated peptide sequence and observed ions are indicated on top of spectra. Singly and doubly charged a, b and y ions, as well as ions corresponding to neutral losses of water and NH3 groups; M, parent ion mass are shown. The inset shows the relative abundance of the monoubiquitinated AVTKFTSS peptide in wild-type (arbitrarily set to 1) and *det1-1* mutant plants. Intensity is calculated as the normalized average integrated MS peak area of the modified peptide from three independent biological replicates for each plant line. Data are represented as mean with 5% confidence interval. More details are given in the Experimental Procedures. (**C**) Immunoblot analysis of chromatin extracts from wild-type and *det1-1* seedlings performed with the indicated antibodies. (**D**) Immunoblot analysis of H2Bub in *det1-1* mutant seedlings, performed as in (**C**). Identity of the H2Bub band is confirmed by co-migration of *hub1-3* mutant chromatin extracts. In (**C–D**), the anti-H2B antibody allows detecting simultaneously core histone H2B (18 kDa) and monoubiquitinated (~24 kDa) H2B forms.

DOI: https://doi.org/10.7554/eLife.37892.002

The following source data and figure supplement are available for figure 1:

**Source data 1.** Detailed identification of differentially modified histone peptides in wild-type and *det1-1* samples.
DOI: https://doi.org/10.7554/eLife.37892.004
**Figure supplement 1.** Mass spectrometry analysis of histone PTM defects in *det1-1* mutant plants.
DOI: https://doi.org/10.7554/eLife.37892.003

H3K36me3 or H3K27me3 could be detected by immunoblot analysis (*Figure 1C*). Instead, complementary immunoblot analyses of euchromatic histone marks detected increased H3K9ac levels and reduction of H3K27ac levels in *det1-1* (*Figure 1C*), which had not been found by mass spectrometry.

Given the affinity of DET1 for histone H2B (*Benvenuto et al., 2002*) and its implication in the ubiquitination pathway (*Osterlund et al., 2000*; *Yanagawa et al., 2004*), we focused on the reduction of H2Bub levels in *det1-1* seedlings. We first assessed on which loci DET1 impacted the genomic landscape of this chromatin mark by conducting a H2Bub ChIP-seq analysis of wild-type and

*det1-1* mutant seedlings grown under dark or light conditions. To allow for quantitative comparisons of differential enrichments in samples displaying drastic differences in H2Bub abundance, 3% Drosophila chromatin was spiked in each input, and *ad hoc* normalization of the Input/IP signals was performed for each sample individually (modified from the original ChIP with reference exogenous genome ChIP-Rx protocol from (*Orlando et al., 2014*); see Methods and Materials). Consistent with our previous findings obtained by ChIP-chip (*Bourbousse et al., 2012*; *Roudier et al., 2011*), H2Bub domains were largely restricted to the transcribed regions. While a set of ~6900 genes consistently bear a H2Bub peak in all four samples, the number of marked genes in *det1-1* compared to WT plants was 19% lower in the light (2134 fewer genes) and 26% lower in the dark (2577 fewer genes) (*Figure 2A*). Quantification using Rx scaling factors unveiled a severe decrease of H2Bub average level over the set of marked genes in *det1-1* seedlings, which was similarly observed in both light conditions (*Figure 2B–C*). Identification of the H2Bub-marked genes that are differentially ubiquitinated (DUGs) further showed that H2Bub is decreased over almost all genes in *det1-1* plants, regardless of the light condition (*Figure 2D*).

As previously reported (*Schroeder et al., 2002*), RNA-seq analyses performed on the same samples showed that *det1-1* mutation has a broad effect on gene expression, to a large extent mimicking the effect of light signals on the transcriptome in dark conditions (*Figure 2—figure supplement 1A*). In both dark and light conditions, gene expression misregulation occurred in a globally hypo-ubiquitinated H2B landscape in *det1-1* plants (*Figure 2E*). Small variations in H2Bub levels between dark and light-grown samples correlating with transcript levels could be detected in *det1-1* plants, although to a much lower extent than in wild-type plants (*Figure 2—figure supplement 1B*). All together, these observations indicate that DET1 impacts the whole H2Bub landscape, without apparent gene specificity and without clear functional impact on steady state transcript profiles.

## Light triggers a global increase of H2Bub enrichment

In line with the role of H2Bub in the induction of light-responsive genes (*Bourbousse et al., 2012*), variations in H2Bub levels correlating with transcript levels between dark and light conditions could be detected over hundreds of genes in WT plants (*Figure 2—figure supplement 1B*). For example, the upregulated *HCF173* gene involved in the Photosystem II biogenesis under light conditions (*Schult et al., 2007*) was detected as being hyper-ubiquitinated (hyper-Differentially Ubiquitinated Gene or hyper-DUG) (*Figure 2—source data 1*) and visually displayed higher H2Bub levels in WT (L) than in WT (D) samples (*Figure 2F*). Vice versa, the light-repressed *HCAR* gene involved in chlorophyll breakdown in darkness (*Sakuraba et al., 2013*) was detected as a hypo-DUG, and visually displays reduced H2Bub levels over its coding sequence (*Figure 2F*).

More unexpectedly, comparison of light- and dark-grown WT plants further unveiled a global tendency in higher H2Bub chromatin enrichment in the light than in the dark condition. As compared to the dark condition, in the light i) the number of H2Bub-marked genes was 12% higher (*Figure 2A*), ii) H2Bub average enrichment was higher in meta-plots representing all 11,921 H2Bub-marked genes (*Figure 2C*) and accordingly, iii) hyper-DUGs in the light were 10 times more numerous than hypo-DUGs (*Figure 2G*). Correlation analysis of read counts in all replicates confirmed a dual effect of both light and *det1-1* mutation on global H2Bub dynamics between dark and light conditions (*Figure 2—figure supplement 1C*).

Comparison of relative transcript levels in dark- and light-grown WT seedlings showed that an equivalent number of genes are up- and down-regulated by light (*Figure 2—figure supplement 1A*), with similar amplitude in transcript level variations (*Figure 2—figure supplement 1D*). Consequently, on their own, relative changes in gene expression do not seem to explain the global tendency for higher H2Bub enrichment in the light *versus* dark condition. These findings unveil that H2Bub is globally more abundant on genes in photomophogenic than in etiolated seedlings, a light-driven dynamic that is lost upon *DET1* loss-of-function. Considering its genome-wide property, the process might relate to variations in H2Bub homeostasis.

## DDA1 associates with SGF11, a potential SAGA-like H2Bub de-ubiquitination component

Considering its implication in the ubiquitination pathway (*Chory, 2010*; *Lau and Deng, 2012*), we suspected that DET1 might exert a direct regulatory role on H2B ubiquitination. Attempts to detect

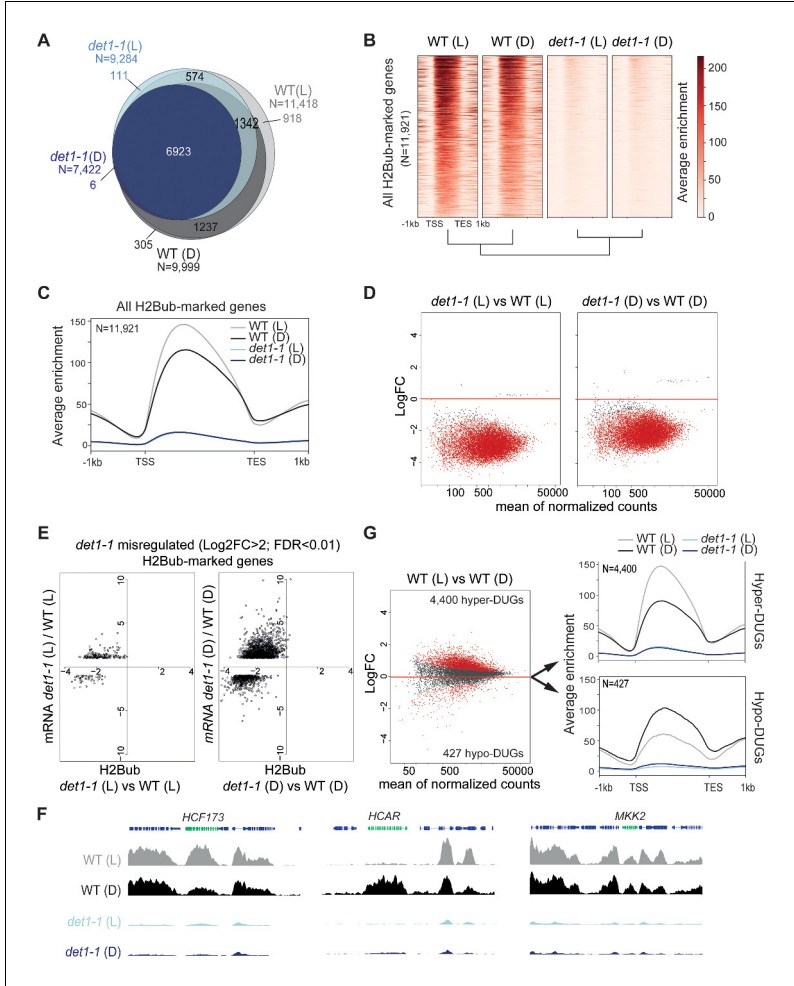

**Figure 2.** Light and DET1 massively influence H2Bub enrichment over protein-coding genes. (**A**) Number of H2Bub-marked genes in 5-day-old wild-type (WT) and *det1-1* seedlings grown under light (**L**) or dark (**D**) conditions. A gene was considered as being marked when overlapping a H2Bub peak in each of the two biological replicates. (**B**) Heatmap showing dramatically low H2Bub levels over H2Bub-marked genes in *det1-1* seedlings. Genes are ranked from top to down according to H2Bub level in the WT (**L**) sample. (**C**) Metagene plot of H2Bub distribution over the coding regions of the 11,921 H2Bub-marked genes in the four sample types. (**D**) Identification of differentially ubiquitinated genes (DUGs) by Rx-normalized DESeq2 analysis (FDR < 0.01) shows that *det1-1* mutation triggers low H2Bub levels over almost all genes. Dots represent the full set of genes displaying a H2Bub domain (according to MACS2 peak detection, see Materials and methods) in at least one sample type. Red dots correspond to differentially ubiquitinated genes (DUGs; FDR < 0.01). (**E**) In both dark and light conditions, massive misregulation of gene expression occurred in a globally hypo-ubiquitinated H2B landscape in *det1-1* plants. Scatter plots show the correspondence between H2Bub and gene expression changes between WT and *det1-1* plants on *det1-1* misregulated genes in light (left) and dark (right) conditions. The x-axis shows Log2 fold-changes (FC) of H2Bub levels as determined by Rx-normalized DESeq2 analysis (FDR < 0.01). The y-axis shows expression Log2 fold changes of as determined by DESeq2 (>2 fold variation, FDR < 0.01). Only genes bearing a significant H2Bub domain according to MACS2 peak detection are shown. (**F**) Genome-browser snapshots showing H2Bub profiles in WT and *det1-1* seedlings grown in light or dark conditions. *MAP KINASE KINASE 2* (*MKK2*) is invariably expressed and marked by H2Bub under dark and light conditions. *HIGH CHLOROPHYLL FLUORESCENCE* (*HCF173*) is a hyper-DUG (Log2FC = 1.3) induced by light (Log2FC = 3.7) that was previously shown to require HUB activity for optimal inducibility by light (***Bourbousse et al., 2012***). In contrast, *7-HYDROXYMETHYL CHLOROPHYLL A* (*HCAR*) is a hypo-DUG (Log2FC = −3.4) repressed by light (Log2FC = −0.7). (**G**) Similar analyses as in (**C**) and (**D**) showing a tendency towards higher H2Bub enrichment in light than in dark condition in wild-type plants. The 4400 hyper-DUGs and 427 hypo-DUGs in WT (**L**) versus WT (**D**) samples and their corresponding H2Bub meta-profile are shown. In (**C**) to (**G**) H2Bub levels were scaled according to ChIP-Rx normalization factors calculated for each sample type to adjust for quantitative IP/Input variations in

*Figure 2 continued on next page*

*Figure 2 continued*

H2Bub enrichment over the genome. In all analyses, each sample type is the mean of two independent biological replicates.

DOI: https://doi.org/10.7554/eLife.37892.005

The following source data, source code and figure supplements are available for figure 2:

**Source data 1.** Gene lists summarizing ChIP-Rx analysis of H2Bub levels in WT and *det1-1* seedlings grown under dark or light conditions.

DOI: https://doi.org/10.7554/eLife.37892.007

**Source data 2.** RNA-seq identification of genes differentially expressed in WT, *det1-1*, *ubp22-1* and *det1-1ubp22-1* mutant lines grown under dark or light conditions using DESeq2.

DOI: https://doi.org/10.7554/eLife.37892.008

**Source code 1.** ChIP differential analyses.

DOI: https://doi.org/10.7554/eLife.37892.009

**Source code 2.** RNA-seq differential analyses.

DOI: https://doi.org/10.7554/eLife.37892.010

**Figure supplement 1.** Light and DET1 massively influence H2Bub enrichment over protein-coding genes.

DOI: https://doi.org/10.7554/eLife.37892.006

---

physical interactions of either DET1 or of its tightly associated DDB1 adaptor protein with HUB1 or HUB2 by yeast-two-hybrid (Y2H) and by co-purification were unsuccessful (data not shown). However, a Y2H screen using DDA1 as bait reproducibly detected the coding sequence of *AT5G58575*, a gene recently identified as the closest Arabidopsis ortholog of yeast *Sgf11* (*Moraga and Aquea, 2015*; *Srivastava et al., 2015*). *AT5G58575*, hereafter referred to as *SGF11*, encodes a protein with 31% and 19% sequence similarity with the SAGA DUB module component Sgf11/ATXN7L3 from *S. cerevisiae* and human, respectively (*Figure 3—figure supplement 1*). Reciprocal Y2H assays using a 3AT chase control confirmed the specific association of DDA1 with SGF11 (*Figure 3A*).

Bimolecular fluorescence complementation (BiFC) assays in *Nicotiana benthamiana* leaves further showed that SGF11 associates with DDA1 in plant nuclei (*Figure 3B*) but neither with DET1 nor with the other two C3D components COP10 and DDB1a (*Figure 4—figure supplement 1*). Finally, DDA1 and SGF11 were found to associate in Arabidopsis extracts using semi-in vivo maltose-binding protein (MBP) fusion pull-down assays (*Figure 3C*). In this assay, recombinant MBP-SGF11 fusion protein purified from *Escherichia coli* cells was used to pull down DDA1 from extracts of DDA1-GFP overexpressing plants. Similar MBP assays showed that DET1 and SGF11 are also associated, although with a lower efficiency (*Figure 3D*). We conclude from these analyses that DDA1 physically associates with SGF11, a link that potentially allows bridging the C3D complex to a SAGA-like H2Bub deubiquitination activity.

## SGF11 makes part of a trimeric H2Bub DUB module

Given the lack of knowledge on SAGA DUBm composition, structure and activity in the plant kingdom, we carried out Tandem Affinity Purification (TAP) analysis of SGF11 constitutively expressed in Arabidopsis cell cultures grown under either light or dark conditions (*Figure 4A*). This consistently identified peptides from four proteins in addition to SGF11 itself: two cytosolic TONNEAU1A/B proteins (*Traas et al., 1987*) as well as AT3G27100 and AT5G10790, two orthologous proteins of the yeast/Drosophila and human DUBm subunits SUS1/ENY2 and UBP22/USP22 proteins, respectively. SUS1 already referring to SUSPENSOR-1 in Arabidopsis (www.tair.org), these were named ENY2 and UBP22. Reciprocal TAP analyses using UBP22 as bait identified only SGF11 and ENY2 (*Figure 4—source data 1*), thus confirming robust and specific co-purification of these three proteins. Altogether, our TAP analyses allowed the co-purification of three of the four known SAGA DUBm subunits, SGF73 being conspicuously absent. Noteworthy, the Arabidopsis genome apparently encodes no orthologous proteins of the yeast/human SGF73/ATXN7 (*Moraga and Aquea, 2015*; *Srivastava et al., 2015*), a subunit that is required for stabilizing and bridging DUBm to SAGA in other species (*Durand et al., 2014*). Our attempts to identify proteins displaying partial or full-length similarities to *S. cerevisiae* SGF73 primary or secondary structure in the genomes of green plant and algal lineages have also been unsuccessful (see Methods and Materials).

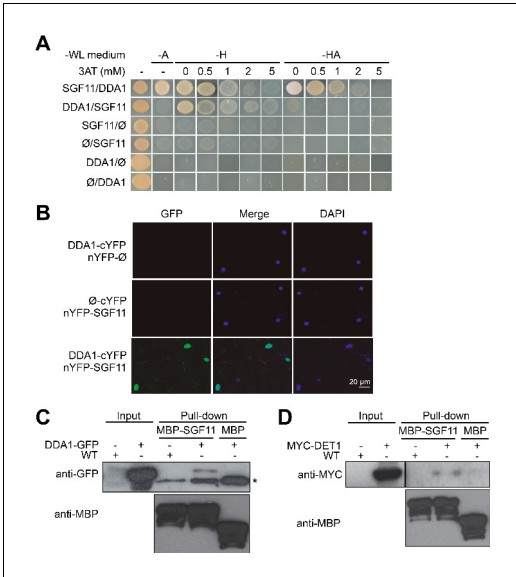

**Figure 3.** DDA1 C3D subunit physically associates with SGF11. (**A**) Yeast-two-hybrid validation of DDA1-SFG11 interaction identified by Y2H screen. (**B**) BiFC analysis of DDA1 physical interaction with SGF11 *in planta*. Live fluorescence imaging 72 hr after dual transfection of Nicotiana leaves by plasmids expressing the indicated translational fusions. Merged images of GFP (left) and DAPI (right) channels are shown (middle). (**C–D**) DDA1 and DET1 associate with the SGF11 in plant extracts. Pull-down assays were performed using total protein extracts prepared from 7-d-old seedlings expressing DDA1-GFP (**C**) or MYC-DET1 (**D**) protein fusions with MBP-SGF11. Protein extracts (Input; 2% of the total amount used) and pull-down proteins were subjected to immunoblot analysis with anti-GFP, anti-MYC and anti-MBP to detect the corresponding fusion proteins. In (**C**), an asterisk indicates the position of an anti-GFP cross-reacting signal.

DOI: https://doi.org/10.7554/eLife.37892.011

The following figure supplement is available for figure 3:

**Figure supplement 1.** Amino-acid sequence alignments of Arabidopsis, *S.cerevisiae* and human SGF11, ENY2/SUS and UBP22/Ubp8/USP22 proteins.
DOI: https://doi.org/10.7554/eLife.37892.012

Y2H and BiFC experiments were conducted to assess the mode of association between SGF11, ENY2 and UBP22. As observed in *S. cerevisiae* (*Köhler et al., 2006*), both analyses showed robust interactions of SGF11 with ENY2 and UBP22, but not between ENY2 and UBP22 (*Figure 4B–C*). To access how SGF11 might bridge ENY2 to UBP22 in the absence of an SGF73/ATXN7 ortholog, we modeled the tridimensional structure of the Arabidopsis DUBm based on their sequence homology with proteins of known structure. The best confidence model displayed high similarity to the yeast DUBm X-ray structure made of the four Sgf11, Ubp8, Sus1 and Sgf73 proteins (*Samara et al., 2010*) (*Figure 4D*). This representation shows conservation of the UBP22 Zn-finger domains positioning, in particular that potentially forming the conserved assembly lobe by encapsulating the long SGF11 amino-terminal α-helix with the central part of the small ENY2 protein (*Köhler et al., 2010*; *Samara et al., 2010*). The small ENY2 protein forms an interface between the UBP22 C-terminal domain and the long SGF11 α-helix, which may re-enforce the pairwise protein interactions in the DUBm (*Figure 4C*). A catalytic lobe oriented away from the assembly lobe can also be formed by the predicted ubiquitin-protease domain of UBP22. SGF11 C-terminal zinc-finger potentially allows interaction with a nucleosome and the ubiquitin moiety of H2Bub as shown in other species (*Durand et al., 2014*; *Samara et al., 2012*). Interestingly, long plant-specific amino- and carboxy-terminal extensions may protrude out from the conserved structural domains, possibly linking this core structure to other proteins or replacing the SAGA bridging-function of Sgf73.

Arabidopsis transgenic lines stably expressing GFP-tagged SGF11, ENY2 or UBP22 were generated and used to investigate their subcellular localization, confirming that all proteins are nuclear-localized (*Figure 4—figure supplement 2A*). In agreement with the conserved function of SAGA in RNA Pol II transcription regulation, confocal imaging after anti-GFP immunolabeling further showed that SGF11 and UBP22 displayed punctuated signals in the euchromatin and are visibly excluded from densely DAPI-stained heterochromatic chromocenters and from the nucleolus. Euchromatic localization was also observed for DDA1-GFP and MYC-DET1 proteins (*Figure 4—figure supplement 2B*). By contrast, GFP-ENY2 signals were reproducibly enriched both in the euchromatic and heterochromatic compartments, sometimes overlapping with nucleolus-associated chromocenters (*Figure 4—figure supplement 2B*). Though, at this stage we also cannot exclude that GFP-ENY2 signal in heterochromatic domains might artifactually result from its overexpression, Arabidopsis ENY2 might therefore display another activity independently from SGF11 and UBP22 in Arabidopsis. As further supported by recent FRAP assays (*Pfab et al., 2018*), ENY2 function potentially differs from its metazoan counterparts that are enriched at the nuclear periphery (*Rodríguez-Navarro et al., 2004*).

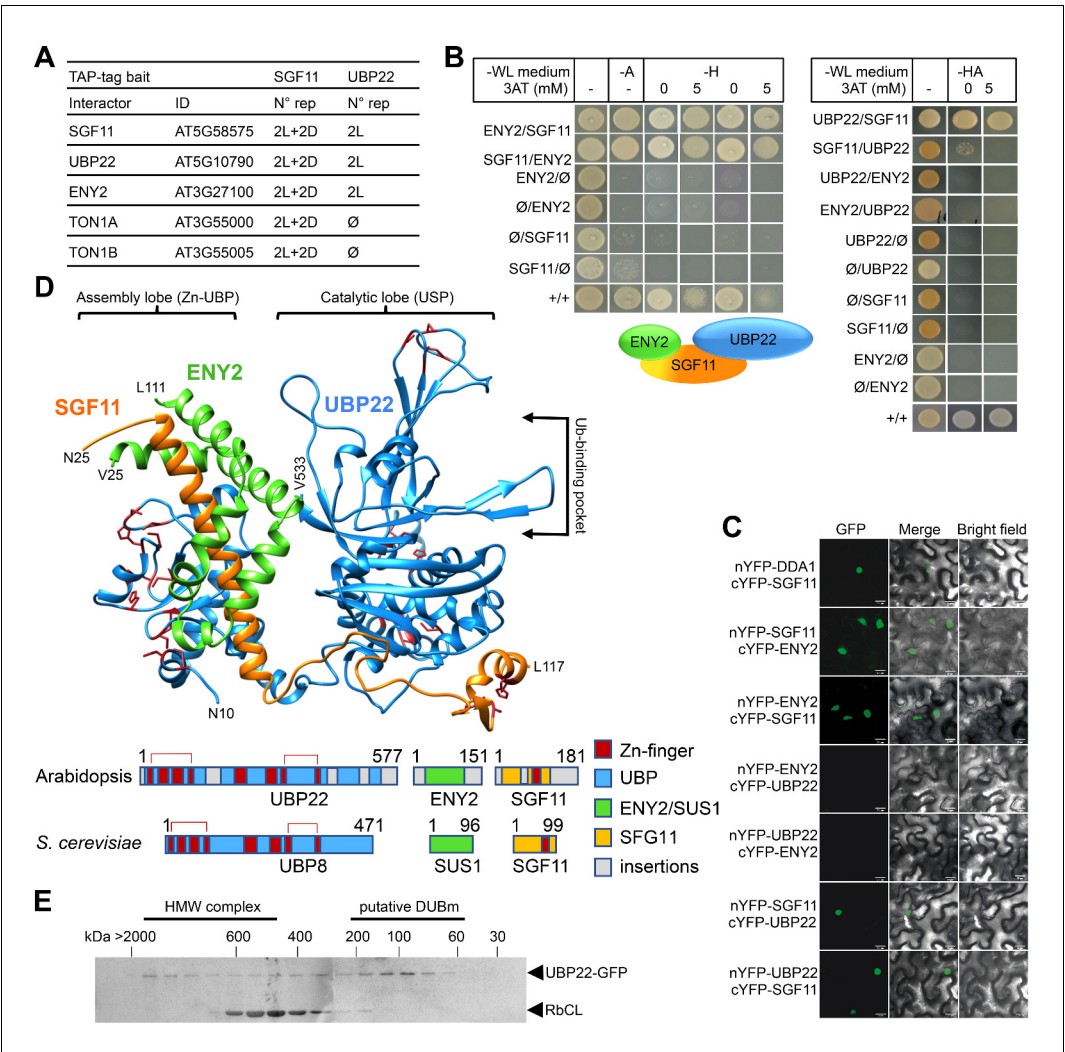

**Figure 4.** SGF11 physically associates with ENY2 and UBP22 to form an Ubp8-like DUBm. (**A**). TAP identification of UBP22 and ENY2 as SGF11-associated proteins. The table summarizes the whole set of detected proteins in each assay, including two independent replicates of dark (D) or light (L) grown cells using SGF11 as bait, and two independent biological replicates of light-grown cells when using UBP22 as a bait. (**B**) Yeast two-hybrid analysis of physical association between SGF11, UBP22 and ENY2 showing dual association of SGF11 with the two other DUBm components. (**C**) BiFC analysis of SGF11 interaction with UBP22 and ENY2 *in planta*. Merged images of GFP (left) and bright field (right) channels are shown (middle). Ø, empty vector. (**D**) Top, modeled structure of the *Arabidopsis* UBP22-SGF11-ENY2 complex. Bottom, representation of domain similarities in the Arabidopsis and *S. cerevisiae* proteins (colored as indicated, gray areas reveal insertions that could not be modeled). The position of predicted Zn-finger domains in UBP22 and SGF11 is indicated in dark red in both panels. Red brackets above the schematized sequences link domains involved in the same Zn-finger. (**E**) Size-exclusion chromatography analysis of UBP22-GFP complex size in protein extracts from *UBP22::UBP22-GFP* Arabidopsis seedlings. Fractions were analyzed by anti-GFP immunoblot. Highly abundant Ribulose bisphosphate carboxylase Large Chain (RbCL) bands were also non-specifically detected. The size of molecular-weight standard eluted in the same conditions is shown.

DOI: https://doi.org/10.7554/eLife.37892.013

The following source data and figure supplements are available for figure 4:

**Source data 1.** Detailed TAP protein identification.
DOI: https://doi.org/10.7554/eLife.37892.016
**Source data 2.** pdb of the modeled At-UBP-ENY2-SGF11 complex structure.
DOI: https://doi.org/10.7554/eLife.37892.017
**Figure supplement 1.** SGF11 in vivo association with the C3D complex occurs through direct interaction with DDA1.
DOI: https://doi.org/10.7554/eLife.37892.014
**Figure supplement 2.** DET1, DDA1 and the UBP22 DUBm subunits are enriched in euchromatin whereas ENY2 is also found in heterochromatic foci.
DOI: https://doi.org/10.7554/eLife.37892.015

Taken together, these observations unveil that Arabidopsis SGF11, ENY2 and UBP22 associate *in planta* and potentially form a trimeric DUB module in the absence of an Sgf73/ATXN7 ortholog. To assess the stable accumulation of such a complex, a size-exclusion chromatography analysis of soluble protein fractions from the *UBP22::UBP22-GFP* plant line was performed. The ~100 kDa GFP-UBP22 protein was largely part of a 100–150 kDa complex, a size compatible with a tripartite association with the 20 kDa SGF11 and 13 kDa ENY2 proteins (*Figure 4E*). UBP22-GFP could also be detected (albeit to a much lower extent) in higher molecular weight fractions, which could correspond to SAGA-like complex size.

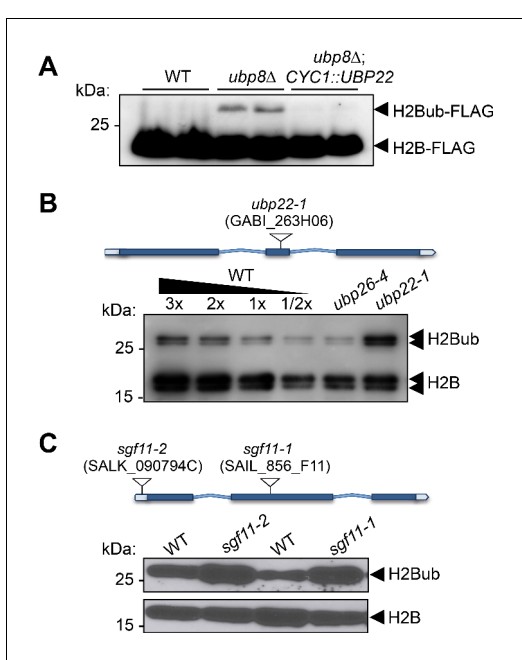

**Figure 5.** UBP22 is a functional homolog of the *S. cerevisiae* Ubp8 required for histone H2B deubiquitination in Arabidopsis. (**A**) Complementation of a *S. cerevisiae ubp8Δ* mutant line (*Henry et al., 2003*) by expression of the Arabidopsis *UBP22* CDS under the *Cyc1* promoter. For each line, FLAG-tagged histone H2B was detected by immunoblot analysis of whole cell extracts from two different yeast transformed colonies. (**B**) Anti-H2B immunoblot analyses of H2Bub in *ubp22-1* and *ubp26-4* mutant seedlings indicate that UBP22 has a more prevalent role than UBP26 in H2Bub removal. Core histone H2B (18 kDa) forms and a doublet corresponding to monoubiquitinated H2B forms (~26 kDa) are detected in chromatin-enriched wild-type extracts. (**C**) Similarly to *ubp22-1*, *SGF11* loss of function triggers a large increase in H2Bub level. In (**B**) and (**C**), the position and ID of the different T-DNAs are depicted on the gene models.
DOI: https://doi.org/10.7554/eLife.37892.018

The following source data and figure supplement are available for figure 5:

**Source data 1.** original blot of data in (**B**).
DOI: https://doi.org/10.7554/eLife.37892.020
**Figure supplement 1.** Functional analysis of the *ubp22-1*, *sgf11-1* and *sgf11-2* mutant lines.
DOI: https://doi.org/10.7554/eLife.37892.019

## UBP22 is a major H2Bub deubiquitinase

To determine whether UBP22 displays H2Bub deubiquitination activity, we conducted a complementation assay of an *S. cerevisiae* Ubp8 knockout strain, a loss-of-function that induces aberrantly high H2Bub levels (*Gardner et al., 2005*; *Henry et al., 2003*). Expression of the full-length *UBP22* coding sequence under the control of the yeast *CYC1* promoter successfully restored normal H2Bub levels in independently transformed colonies (*Figure 5A*). UBP22 can therefore replace Ubp8 activity for H2Bub hydrolysis in the context of a *bona fide* SAGA complex.

To further test the influence of the Arabidopsis DUBm on H2Bub in plants, we obtained T-DNA insertion lines for *UBP22* and *SGF11*, hereafter referred to as *ubp22-1*, *sgf11-1* and *sgf11-2*. All homozygous plant lines were viable and displayed no obvious developmental phenotypes under standard laboratory growth conditions (*Figure 5—figure supplement 1A*). Immunoblot analyses showed a robust increase of H2Bub levels in each of these loss-of-function lines as compared to their wild-type counterparts, resulting in 2–3 fold more monoubiquitination of the histone H2B pool than in wild-type plants (*Figure 5B–C*). Re-introduction of a *UBP22* or *SGF11* functional coding sequence successfully restored low H2Bub levels in *ubp22-1* and *sgf11-2* plants, respectively (*Figure 5—figure supplement 1B*). As shown in *Figure 5B*, comparison with the previously identified histone H2Bub deubiquitination mutant line *ubp26-4* (*Schmitz et al., 2009*), showed a pronounced effect in *ubp22-1*. These observations indicate that UBP22 constitutes a major determinant of histone H2Bub deubiquitination and is necessary to maintain normal levels of H2Bub in Arabidopsis.

## Light and DET1 control the proteolytic degradation of the SGF11 DUBm subunit

Considering the dual effects of light and of *DET1* loss-of-function on H2Bub levels on one hand, and the physical association of DDA1 with the

DUBm component SGF11 on the other, we tested whether SGF11 protein abundance was influenced by light conditions. Analysis of GFP-SGF11 expressed under the control of a 35S promoter (*35S:: GFP-SGF11* plants) indeed showed lower levels in protein extracts from dark-grown than in light-grown seedlings (*Figure 6A*). Pre-incubation of the plants with the MG132 proteasome inhibitor largely increased GFP-SGF11 accumulation to similar levels in both conditions. Similar effects were observed for the ENY2 and UBP22 proteins (*Figure 6A*), showing that abundance of all DUBm subunits is tightly controlled. Affinity purification of protein extracts from *35S::MYC-SGF11* seedlings with a ubiquitin-affinity resin showed that SGF11 proteins are poly-ubiquitinated under both dark and light conditions, with a more pronounced effect in darkness (*Figure 6B*). We concluded that stability of SGF11 is controlled by a proteasomal activity with a more prevalent effect in darkness.

Given the established role of the C3D complex in promoting protein degradation in darkness, we further tested whether the effect of light on SGF11 stability is mediated by DET1 after introgression of the *det1-1* allele in the *35S:: MYC-SGF11* and *35S::GFP-UBP22* transgenic lines. Levels of both MYC-SGF11 and GFP-UBP22 were higher in *det1-1* than in wild-type plants in both conditions, indicating that DET1 is indeed required for SGF11 and UBP22 proteolytic degradation (*Figure 6C*). To test for a dynamic regulation of SGF11 and UBP22 stability by light signaling, the *35S::MYC-SGF11* and *35S::GFP-UBP22* transgenic lines were submitted to dark-to-light and light-to-dark switches. Steady-state levels of the constitutively expressed recombinant MYC-SGF11 and GFP-UBP22 proteins exhibited dramatic changes during the transitions, which were already visible after 24 hr upon the transitions (*Figure 6D*, top panels). Furthermore, these changes were robustly dampened in the *det1-1* genetic background (*Figure 6D*, bottom panels). Taken together, these observations indicate that stability of the DUBm complex is regulated by the light signaling factor DET1.

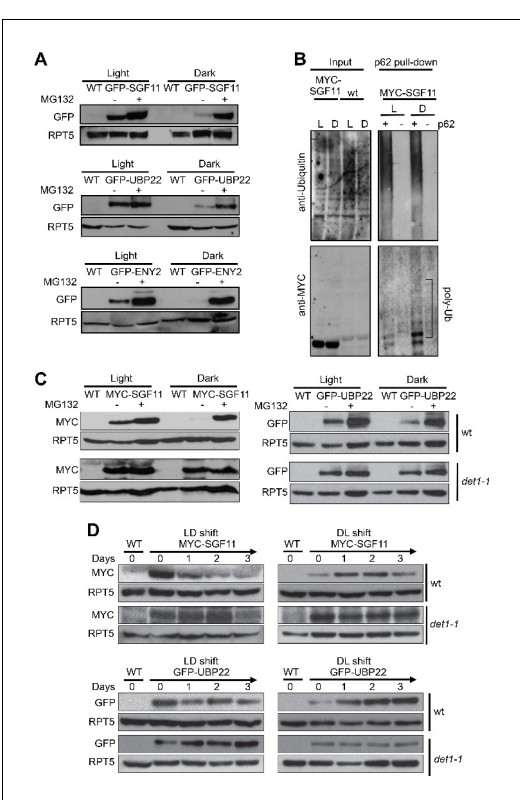

**Figure 6.** Light signaling controls the proteolytic degradation of the UBP22, SGF11 and ENY2 DUBm subunits. (**A**). UBP22, SGF11 and ENY2 DUBm components are degraded by the proteasome preferentially in darkness. Immunoblot analysis of GFP-tagged DUBm proteins in seedlings grown under continuous light or dark conditions. Prior to plant harvesting, seedlings were treated (+) or not (-) with 50 µM MG132 for 12 hr. (**B**) SGF11 is poly-ubiquitinated in vivo. MYC-SGF11 protein extracts from light (L)- and dark (D)-grown seedlings, treated (+) or not (-) with 50 µM MG132 for 12 hr, were incubated with p62 resin or with agarose resin as negative control. Ubiquitinated proteins are detected using anti-Ubiquitin antibody. Anti-MYC allows detection of MYC-SGF11 and its polyubiquitinated isoforms. Polyubiquitinated MYC-SGF11 signals are indicated (poly-Ub). (**C**) DET1 promotes the degradation of SGF11 and UBP22 preferentially in darkness. Analysis of the abundance of MYC-SGF11 and GFP-UBP22 proteins in wild-type and *det1-1* seedlings grown under dark or light conditions. Prior to plant harvesting, seedlings were treated (+) or not (-) with 50 µM MG132 for 12 hr. (**D**) DET1 promotes UBP22 and SGF11 protein degradation upon a light-to-dark shift. In (**A-D**), immunoblots were performed on whole-cell protein extracts from seedlings grown under the indicated conditions.

DOI: https://doi.org/10.7554/eLife.37892.021

## DET1 antagonizes DUBm activity on H2Bub removal

Our findings point towards a model in which DET1 influences H2Bub homeostasis by opposing DUBm activity. To examine this possible antagonistic influence, we crossed the *det1-1* and *ubp22-1* lines and probed global H2Bub levels in double mutant plants. Immunoblot analyses indeed showed increased H2Bub levels in *det1-1ubp22-1* as compared to *det1-1* single mutant seedlings. Hence, loss of H2Bub triggered by

*det1-1* is partially suppressed in the absence of a functional DUBm (*Figure 7A*).

To provide a more functional assessment on the antagonistic activities of DET1 and DUBm, we determined the photomorphogenic phenotype of *ubp22-1* and *det1-1ubp22-1* seedlings by measuring hypocotyl length upon growth in darkness. Again, the *ubp22-1* mutation partially suppressed the constitutive photomorphogenic phenotype of *det1-1* seedlings (*Chory et al., 1989*), whereas *UBP22* overexpression on its own did not affect hypocotyl elongation (*Figure 7B*). Secondly, we determined the transcriptome profile of *det1-1ubp22-1* seedlings and compared it with those of WT, *det1-1* and *ubp22-1* plants. As reported for the yeast *ubp8Δ* strain (*Gardner et al., 2005*; *Lenstra et al., 2011*), differential analyses of gene expression showed that *UBP22* loss-of-function only marginally affects transcript levels - with just 26 genes being misregulated in *ubp22-1* as compared to WT plants (*Figure 7C*). In line with the hypothesis of C3D opposing DUBm activity, *det1-1ubp22-1* double mutant plants displayed a partially suppressed transcriptome profile as compared to *det1-1* plants, the expression of about half (53%) of the genes misregulated in *det1-1* seedlings being restored by *UBP22* loss-of-function (*Figure 7C*).

Our conclusions on C3D function therefore predict that over-accumulation of the DUBm in *det1-1* plants is responsible, at least partially, for the reduction of H2Bub levels in *det1-1* plants. Independent transgenic lines in which *UBP22-GFP* is overexpressed (*wt/UBP22-GFP#OE1 and #OE2*) indeed confirm that UBP22 is a limiting factor in H2Bub removal (*Figure 7E*). Taken together, these findings indicate that DET1-mediated control of DUBm proteolytic degradation represents a previously unidentified mechanism to adjust H2Bub levels, which can mediate light-dependent control on H2Bub homeostasis over the genome.

## Discussion

### Photomorphogenesis combines drastic changes of chromatin organization, transcriptional activity and H2Bub enrichment over the genome

Many plant cells are subjected to drastic transcriptomic and metabolic changes during the photomorphogenic transition, a process also involving nuclear expansion, heterochromatin condensation, and variations of nucleosomal accessibility over a large repertoire of genes (*Bourbousse et al., 2015*; *Peschke and Kretsch, 2011*; *Sullivan et al., 2014*). Hundreds of genes are differentially marked by H2Bub during de-etiolation, a phenomenon that we could link to co-occurring gene expression changes (*Bourbousse et al., 2012*), many of which undergo pioneering rounds of transcription during photomorphogenesis (reviewed in *Barneche et al., 2014*; *Perrella and Kaiserli, 2016*). Accordingly, dual detection of non-phosphorylated and Ser-5 phosphorylated RNA Pol II CTD in individual cotyledon nuclei has shown that the proportion of RNA Polymerase II molecules engaged in transcription increases more than twice when comparing dark and light conditions (*Bourbousse et al., 2015*). The H2Bub ChIP-Rx analysis performed here further reveals that H2Bub abundance is globally controlled during photomorphogenesis. H2Bub levels were significantly higher on many H2Bub-marked genes in the light than in the dark, while only a few genes, usually light-repressed, display an opposite tendency. This extra layer of information unveils the existence of a control on H2Bub homeostasis by light signaling. Interestingly, the SAGA DUBm mediates the regulation of genes involved in axon targeting and neuronal connectivity in the Drosophila visual system (*Weake et al., 2008*), indicating that functional links between H2Bub deubiquitination and light-related functions exist in distant eukaryotic species.

We further report that the photomorphogenesis regulator DET1 is required for shaping the H2Bub landscape. *DET1* loss-of-function triggers a decrease of H2Bub level throughout almost all genes, a defect that we found to be genetically linked to *UBP22* function. This protein was found to be part of an atypical SAGA-like DUB module whose abundance is regulated by DET1 during dark/light transitions. Taken together, these observations indicate that an active mechanism regulates the availability of the Arabidopsis DUBm, in which DET1 is directly involved. Indeed, SGF11 was found to physically associate with the C3D component DDA1 in yeast two-hybrid experiments and *in planta*. SGF11 is subjected to DET1-dependent poly-ubiquitination and proteolytic degradation, with a more pronounced effect in the dark. Stability, and hence cellular availability of Arabidopsis DUBm is directly controlled by light signaling.

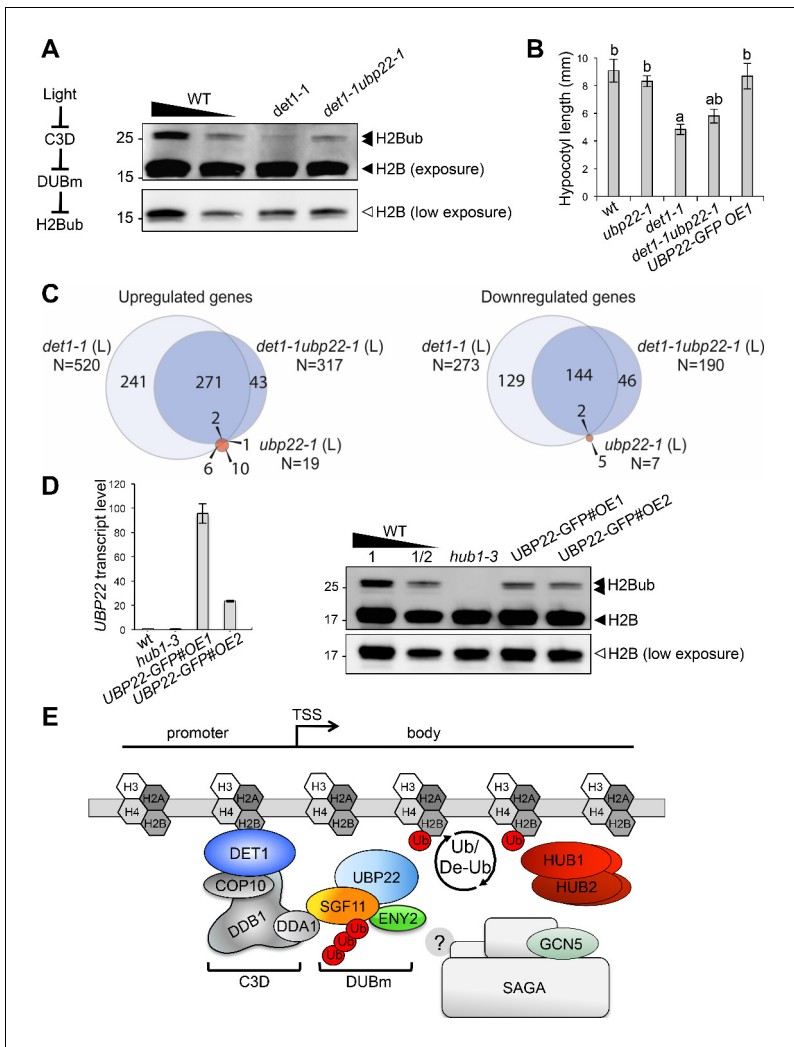

**Figure 7.** DET1 controls histone H2B monoubiquitination levels by opposing DUBm activity. (**A**) *UBP22* loss of function largely suppresses the loss of H2Bub in *det1-1* seedlings. Histone H2B (18 kDa) and its monoubiquitinated H2B (~26 kDa) forms are detected with an anti-H2B antibody in immunoblot analysis of chromatin extracts from seedlings with the indicated genotypes. A scheme for the regulatory effects of light, C3D and DUBm on H2B ubiquitination homeostasis is depicted. (**B**) Partial suppression of *det1-1* constitutive photomorphogenic phenotype by *UBP22* loss-of-function. Columns represent average hypocotyl length of 5-day-old dark-grown seedlings with the indicated genotypes. Error bars represent standard deviation (N ≥ 20). Letters above bars indicate statistically distinct groups (p<0.01, one-way Anova with Tukey-HSD) with respect to wild-type Col-0 (**a**) or to *det1-1* seedlings (**b**). (**C**) RNA-seq analysis of light-grown seedlings show that *det1-1* defects in gene expression are partially suppressed after introgression of the *ubp22-1* mutation, while *UBP22* loss-of-function per se only marginally affects gene expression in wild-type plants. The number of genes correspond to a Log2FC > 1 and a FDR < 0.01. (**D**) *UBP22* overexpression decreases histone H2Bub global level. Left panel, RT-qPCR measurement of *UBP22* transcript levels in overexpression lines #OE1 and #OE2 relative to the wild-type level (arbitrarily set to 1). Right panel, immunodetection of H2Bub levels in chromatin extracts from light-grown seedlings of the indicated genotypes. (**E**) Proposed model depicting the C3D complex activity in regulating histone H2Bub homeostasis through ubiquitin-mediated control of the DUBm stability. UBP22 is a Ubp8 homolog acting with SGF11 and ENY2 in H2Bub deubiquitination, possibly promoting transcription under optimal growth conditions. The C3D complex DDA1 subunit targets SGF11 for degradation, favoring degradation of the DUBm. DET1 affinity for non-acetylated histone H2B (*Benvenuto et al., 2002*) may favor this degradation nearby H2Bub-rich regions. DET1 may further influence histone H2B acetylation status via an unknown mechanism, possibly involving the DUBm/SAGA activities or independent pathways. In the absence of *Sgf73* homolog in plants, the DUBm may act independently from SAGA.

DOI: https://doi.org/10.7554/eLife.37892.022

Our observation of coincidentally low levels of DUB subunits and of H2Bub over the genome in dark condition appears counter-intuitive. Dampening of these two antagonistic components also appear to match a partially poised transcriptional status of fully etiolated seedling cells, a property that might be linked in this case to their stalled development. Although causal relationships cannot be established at this stage, these observations point towards a concerted light-triggered reprogramming of H2Bub homeostasis and cellular transcriptional activity in this biological system, and provide a first potential mechanism as to how the cell controls H2Bub homeostasis during cell specialization. Indeed, RNA Pol II elongation constitutes an essential regulatory step of productive transcription (reviewed in *Li et al., 2007*; *Selth et al., 2010*; *Weake and Workman, 2008*) and histone H2B monoubiquitination is intimately linked with global transcriptional elongation capacity in mammalian cells (*Minsky et al., 2008*). Determining whether modulation of H2Bub levels during photomorphogenesis primarily influences a cell's capacity for transcription elongation or other processes such as DNA replication or cell cycle progression will require additional investigation.

## An atypical H2Bub DUB module is targeted by light signaling for ubiquitin-mediated proteolysis

Despite their involvement in basal cellular mechanisms, RNA Pol II-associated chromatin machineries must themselves be controlled to influence the transcriptional capacity. In budding yeast, the proteasomal ATPase Rpt2 interacts with Sgf73 to dissociate the DUBm from SAGA (*Lim et al., 2013*). Being apparently absent from their genome, a *Sgf73*-based regulatory mechanism cannot be at play in plants. Our findings rather unveil a control of the DUBm abundance through targeted proteolysis of the SGF11 subunit. Confirming FRET and yeast two-hybrid assays performed by *Pfab et al. (2018)*, we found direct associations of SGF11 with both UBP22 and ENY2, suggesting that SGF11 is centrally positioned in the DUBm. This property is also visualized on the modeled UBP22 DUBm structures of our two respective studies (*Figure 4D*; *Pfab et al., 2018*). Hence, C3D-mediated proteolytic degradation of SGF11 is expected to efficiently impair the DUBm activity. UBP22 and ENY2 degradation could either be an indirect consequence of SGF11 proteolysis, or might also result from targeted degradation. Like for many other protein complexes structured by pair-wise interactions, depletion of Ubp8, Sgf11 or Sus1 proteins induces a loss of the other members of the DUBm in yeast, and the same might occur in Arabidopsis. This possibility is supported by the observation that reconstitution of the yeast tetrameric DUBm in vitro requires co-expression of Sus1 and Sgf11 (reviewed in *Rodríguez-Navarro, 2009*). Future studies aimed at dissecting Arabidopsis SGF11 protein domains required for the complex stability and activity might help addressing this question.

Given that SGF73/ATXN7 is required for bridging the DUBm to the SAGA core domain in yeast and human (*Köhler et al., 2008*; *Lee et al., 2005*; *Lee et al., 2009*), identification of a functional SAGA-like DUBm in Arabidopsis lacking a potential ortholog raises questions about the complex formation and activity in plants. Size-exclusion chromatography suggests that UBP22 may to some extent accumulate within a high-molecular weight complex (*Figure 4E*), but such higher-order association may either be transient or labile. Indeed, our UBP22 and SGF11 TAP analyses were unsuccessful in identifying DET1/DDA1 or other SAGA components (*Figure 4—source data 1*). Accordingly, recent characterization of Arabidopsis SAGA complex components by mass spectrometry also failed to find robust association with DUBm subunits by *Pfab et al. (2018)*, altogether suggesting that different SAGA complexes might exist in plants. Interestingly, model distribution simulations of the Arabidopsis DUBm indicate that such a complex could stably accumulate independently from SAGA (*Pfab et al., 2018*). Functional independency between the DUBm and other SAGA modules has been proposed in a few species, with the capacity of a trimeric Ubp8–Sgf11–Sus1 complex to dissociate from SAGA (*Köhler et al., 2006*). The capacity of Arabidopsis ENY2-SGF11-UBP22 to form a fully functional module remains to be tested. Characterizing the function of SGF11 and ENY2 amino or carboxy-terminal extra-domains might reveal how functionality is achieved in a trimeric complex. This might also help to understand the relationship between SAGA-dependent histone acetylation activity and H2Bub dynamics during transcription initiation and elongation in plants. Our mass spectrometry and immunoblot analyses allowed the detection of multiple acetylation defects in histone H2B and H3 in *det1-1* plants. These might result from independent functions of the DET1 protein or possibly also from consequences of impaired SAGA activity. Defective histone acetylation in this mutant line could be meaningful given the influence of histone H2B acetylation for DET1 binding (*Benvenuto et al., 2002*). Hence, concerted action of SAGA HAT and

DUBm activity might both impact DET1 affinity for histone H2B, possibly triggering dynamic changes in DET1 association with chromatin.

## A direct role for DET1 on histone H2B

Despite having no detectable DNA binding capacity, DET1 can act as a potent transcriptional repressor acting in cis on light- and circadian-clock regulated genes in plants or when exogenously expressed in yeast cells (*Lau et al., 2011*; *Maxwell et al., 2003*). This property has been long proposed to relate to its association with histone H2B (*Benvenuto et al., 2002*) but thus far no mechanism has been identified. Here we uncover that DET1 is involved, presumably via the physical association of DDA1 with SGF11, in controlling abundance of a chromatin machinery linked to histone H2B and to RNA Pol II activity. In particular, we propose a possible mechanism involving recruitment of the DDA1 C3D subunit on genomic regions where DET1-mediated modulation of DUBm stability controls H2Bub chromatin enrichment (*Figure 7E*). Decreasing the DUBm availability in darkness might represent a way for the plant cell to trigger global changes in this chromatin mark, possibly dampening H2Bub turnover capacity in cells with a low transcriptional status.

This study shows how the availability of transcriptional co-activator machineries can be modulated by an environmental signal. The impact of Arabidopsis DET1 on the DUBm stability adds to the knowledge that CRL4 activities influence H2Bub dynamics, as shown in the context of circadian clock-controlled gene expression in mammals (*Tamayo et al., 2015*) and of DNA repair pathways in *Schizosaccharomyces pombe* and human cells (*Martinez et al., 2001*; *Zeng et al., 2016*). Our findings on plant photomorphogenesis echo situations in which metazoan cells are also subject to drastic changes in cell identity and proliferation. For example, misregulation of H2Bub homeostasis appears to be central in the transcriptional events linked to human oncogenesis, as *USP22* transcript abundance is part of an 11-gene signature that associates with poor cancer prognosis (*Jeusset and McManus, 2017*), hence potentially influencing cell proliferation processes like human DET1 (*Pick et al., 2007*; *Wertz et al., 2004*). Considering the evolutionary conservation of DET1, DDA1 and of the three plant DUBm components from plants to metazoans, future studies might unveil a functional link between DET1 activity and DUBm abundance in other organisms.

# Methods and materials

**Key resources table**

| Reagent type (species) or resource | Designation | Source or reference | Identifiers | Additional information |
|---|---|---|---|---|
| Genetic reagent (*Arabidopsis thaliana*) | *det1-1* | *Chory et al., 1989* | *det1-1* | Col-0; EMS mutation |
| Genetic reagent (*Arabidopsis thaliana*) | *hub1-3* | *Fleury et al., 2007* | GABI_276D08 | Col-0; from NASC collection |
| Genetic reagent (*Arabidopsis thaliana*) | *hub2-2* | *Liu et al., 2007* | SALK_071289 | Col-0; from NASC collection |
| Genetic reagent (*Arabidopsis thaliana*) | *ubp26-4* | *Sridhar et al., 2007* | SALK_024392 | Col-0; from NASC collection |
| Genetic reagent (*Arabidopsis thaliana*) | *ubp22-1* | This study; see Materials and methods | GABI_263H06 | Col-0; from NASC collection |
| Genetic reagent (*Arabidopsis thaliana*) | *sgf11-1* | This study and *Pfab et al., 2018* | SAIL_856_F11 | Col-0; from NASC collection |
| Genetic reagent (*Arabidopsis thaliana*) | *sgf11-2* | This study; see Materials and methods | SALK_090794C | Col-0; from NASC collection |
| Genetic reagent (*Arabidopsis thaliana*) | wt/*35S::MYC-UBIQUITIN* | *Dubin et al., 2008* | *35S::MYC-UBIQUITIN* | Col-0 |
| Genetic reagent (*Arabidopsis thaliana*) | wt/*35S::MYC-DET1* | *Castells et al., 2011* | *35S::MYC-DET1* | Col-0 |
| Genetic reagent (*Arabidopsis thaliana*) | wt/*35S::DDA1-GFP* | *Irigoyen et al., 2014* | *35S::DDA1-GFP* | Col-0 |

*Continued on next page*

*Continued*

| Reagent type (species) or resource | Designation | Source or reference | Identifiers | Additional information |
|---|---|---|---|---|
| Genetic reagent (*Arabidopsis thaliana*) | wt/*35S::GFP-UBP22* | This study; see Materials and methods | *35S::UBP22-GFP* | Col-0 |
| Genetic reagent (*Arabidopsis thaliana*) | wt/*UBP22::UBP22-GFP#OE2* | This study; see Materials and methods | *UBP22-GFP#OE1* | Col-0 |
| Genetic reagent (*Arabidopsis thaliana*) | wt/*UBP22::UBP22-GFP#OE2* | This study; see Materials and methods | *UBP22-GFP#OE2* | Col-0 |
| Genetic reagent (*Arabidopsis thaliana*) | Complemented *ubp22-1* | This study; see Materials and methods | *ubp22-1/UBP22:UBP22-GFP* | Col-0 |
| Genetic reagent (*Arabidopsis thaliana*) | wt/*35S::GFP-SGF11* | This study; see Materials and methods | *sgf11-2/35S::GFP-SGF11* | Col-0 |
| Genetic reagent (*Arabidopsis thaliana*) | wt/*35S::MYC-SGF11* | This study; see Materials and methods | *sgf11-2/35S::MYC-SGF11* | Col-0 |
| Genetic reagent (*Arabidopsis thaliana*) | wt/*35S:GFP-ENY2* | This study; see Materials and methods | *35S:GFP-ENY2* | Col-0 |
| Strain, strain background (*S. cerevisae*) | *ubp8Δ* mutant line | **Gardner et al., 2005** | UCC6392 | ubp8::KanMX |
| Strain, strain background (*S. cerevisae*) | Complemented *ubp8Δ* mutant line | This study | | CYC1::UBP22 Ubp8::KanMX |
| Antibody | Anti-GFP-HRP | Milteny Biotec #130-091-833 | | Immunoblots (1:2,000) |
| Antibody | anti-FLAG M2 | Sigma-Aldrich #F3165 | | Immunoblots (1:2,000) |
| Antibody | anti-HA-HRP | Roche #3F10 | | Immunoblots, dilution 1:1000 |
| Antibody | anti-human H2B | Millipore #07–371 | | Immunoblots (1:2,000) |
| Antibody | anti-rice H2B | **Bourbousse et al., 2012** | | Immunoblots (1:2,000) |
| Antibody | anti-H2Bub | Medimabs #MM-0029 | | Immunoblots (1:1,000) |
| Antibody | anti-H3K4me2 | Millipore #07–030 | | Immunoblots (1:10,000) |
| Antibody | anti-H3K4me3 | Millipore #05–745 | | Immunoblots (1:10,000) |
| Antibody | anti-H3K27me3 | Millipore #07–449 | | Immunoblots (1:10,000) |
| Antibody | anti-H3K36me3 | Millipore #07–353 | | Immunoblots (1:10,000) |
| Antibody | anti-H3ac | Millipore #06–599 | | Immunoblots (1:10,000) |
| Antibody | anti-H3K9ac | Millipore #06–942 | | Immunoblots (1:10,000) |
| Antibody | anti-H3K27ac | Millipore #07–360 | | Immunoblots (1:10,000) |
| Antibody | anti-H4ac | Millipore #06–598 | | Immunoblots (1:10,000) |
| Antibody | anti-H3 | Millipore #05–499 | | Immunoblots (1:10,000) |
| Antibody | anti-RPT5 | Enzo Life Sciences # BML-PW8245 | | Immunoblots (1:10,000) |
| Antibody | anti-Ubiquitin | Enzo Life Sciences # BML-PW0930 | | Immunoblots (1:1,000) |
| Antibody | anti-MYC | Millipore #05–724 | | Cytology (1:200) |

*Continued on next page*

*Continued*

| Reagent type (species) or resource | Designation | Source or reference | Identifiers | Additional information |
|---|---|---|---|---|
| Antibody | anti-GFP | ThermoFisher Scientific #A11122 | | Cytology (1:200) |
| Antibody | Alexa-488 coupled anti-mouse | ThermoFisher Scientific #A11001 | | Cytology (1:200) |
| Antibody | Alexa-488 coupled anti-rabbit | ThermoFisher Scientific #A11008 | | Cytology (1:200) |
| Antibody | anti-H2Bub | Medimabs #MM-0029-P | | ChIP-seq (1:600) |
| Gene (*Arabidopsis thaliana*) | UBP22 | | At5G10790 | |
| Gene (*Arabidopsis thaliana*) | UBP26 | | At3G49600 | |
| Gene (*Arabidopsis thaliana*) | SGF11 | | AT5G58575 | |
| Gene (*Arabidopsis thaliana*) | ENY2 | | AT3G27100 | |
| Gene (*Arabidopsis thaliana*) | DDB1a | | AT4G05420 | |
| Gene (*Arabidopsis thaliana*) | HUB1 | | At2G44950 | |
| Gene (*Arabidopsis thaliana*) | DET1 | | AT4G10180 | |
| Gene (*Arabidopsis thaliana*) | DDA1 | | AT5G41560 | |
| Gene (*Arabidopsis thaliana*) | HCF173 | | AT1G16720 | |
| Gene (*Arabidopsis thaliana*) | HCAR | | AT1G04620 | |
| Gene (*Arabidopsis thaliana*) | MKK2 | | AT4G29810 | |
| Gene (*Arabidopsis thaliana*) | COP10 | | AT3G13550 | |
| Recombinant DNA reagent | pUBP22::UBP22-GFP (plasmid) | This study, see Materials and methods | | Progenitors: PCR *UBP22*; Gateway vector pB7FWG,0 |
| Recombinant DNA reagent | p35S::GFP-UBP22 (plasmid) | This study, see Materials and methods | | Progenitors: PCR *UBP22*; Gateway vector pB7GWF2 |
| Recombinant DNA reagent | p35S::GFP-SGF11 (plasmid) | This study, see Materials and methods | | Progenitors: PCR *SGF11*; Gateway vector pDONR207 |
| Recombinant DNA reagent | p35S::GFP-ENY2 (plasmid) | This study, see Materials and methods | | Progenitors: PCR *ENY2*; Gateway vector pDONR207 |

## Plant growth conditions

Seeds were surface-sterilized, plated on MS agar medium and, unless stated otherwise, cultivated under either a 16 hr/8 hr (23/19°C) light/dark photoperiod (100 $\mu$mol.m$^{-2}$.s$^{-1}$) or constant (light/23°C or dark/23°C) conditions. Some samples were further transferred to white light (100 $\mu$mol.m$^{-2}$.s$^{-1}$) for the indicated duration and seedlings were harvested concomitantly at 4pm (8zt) under a green safe light for RNA or ChIP extractions.

## Plant lines

All plant lines are in the Columbia-0 ecotype background. The *det1-1* (*Chory et al., 1989*), *hub1-3* (*Fleury et al., 2007*), *hub2-2* (*Liu et al., 2007*), *ubp26-4* (*Sridhar et al., 2007*; *Schmitz et al., 2009*), *35S::MYC-ubiquitin* (*Dubin et al., 2008*) and *35S::MYC-DET1* (*Castells et al., 2011*) Arabidopsis lines have been previously described. The *ubp22-1* (GABI_263H06), *sgf11-1* (SAIL_856_F11) and *sgf11-2* (SALK_090794C) insertion lines were obtained from the NASC (*Scholl et al., 2000*). Transgenic line *UBP22::UBP22-GFP* was obtained by cloning the *UBP22* genomic sequence (*At5g10790*) amplified using the UBP22gF and UBP22gnon*R primers into the pB7FWG,0 vector before plant transformation using floral dipping. The *35S::GFP-UBP22* were lines were obtained by inserting *UBP22* coding sequence in the pB7GWF2 vector using the UBP22_FCl and UBP22_RCl primers. Transgenic lines overexpressing *SGF11* and *ENY2* coding sequences were obtained by Pwo PCR amplification of the corresponding full length coding sequences using the SGF11.BF-SGF11.BR and ENY2.BF-ENY2-BR, respectively, and cloned into the pDONR207 plasmid (Invitrogen) using Gateway BP reaction kits (Invitrogen). The resulting CDS were transferred using Gateway LR reaction kits (Invitrogen) to either pGWB6 or pGWB21 (*Nakagawa et al., 2007*) destination vectors for fusion to GFP or 10xMYC repeats, respectively. Agrobacterium-mediated plant transformation was performed by the floral dip method (*Clough and Bent, 1998*).

## Hypocotyl measurements

Seedlings were photographed 5 days after sowing and images were analyzed with the ImageJ software (https://imagej.nih.gov/ij/). Three biological replicates, each consisting of measurements for at least 20 seedlings grown at different times, were analyzed with similar results. Statistical analyses were assessed by one-way ANOVA and Tukey-HSD test (p<0.01).

## Plant protein extraction and immunoblotting

Chromatin-enriched protein fractions were obtained as previously described (*Bourbousse et al., 2012*). Whole cell protein extracts were obtained by thawing grinded material in 1M Tris-HCl pH8, 20% SDS, 15% glycerol, cOmplete EDTA-free protease inhibitor cocktail (Sigma-Aldrich) followed by centrifugation two times 10 min 16,000 g at 4°C. Extraction of plant soluble protein extracts was performed in 50 mM Tris-HCl, pH 7.4, 150 mM NaCl, 10 mM MgCl2, 1 mM phenylmethylsulfonyl fluoride, 0.1% Nonidet P-40 and cOmplete EDTA-free protease inhibitor cocktail (Sigma-Aldrich) followed by centrifugation two times 10 min 16,000 g at 4°C. Protein concentration in the final supernatants was determined using the Bio-Rad Protein Assay kit. Boiled protein samples were separated by 8–14% SDS-PAGE, transferred on Immobilon-P membranes (Millipore) and blotted with the antibodies detailed below.

## Affinity purification of ubiquitinated proteins

Proteins from light- and dark-grown *MYC-SGF11* seedlings treated with 50 mM MG132 were extracted in buffer BI (50 mM Tris-HCl, pH 7.5, 20 mM NaCl, 0.1% Nonidet P-40, and 5 mM ATP, 1 mM PMSF, 50 mM MG132, 10 nM Ub-aldehyde, and 10 mM N-ethylmaleimide, plant protease inhibitor cocktail [Sigma-Aldrich]) before incubation with pre-washed p62 agarose (Enzo Life Sciences) or the agarose alone at 4°C for 4 hr. Beads were washed twice in BI buffer and once with BI buffer supplemented with 200 mM NaCl. Proteins were eluted in SDS loading buffer at 100°C as described previously (*Manzano et al., 2008*). The eluted proteins were separated by SDS-PAGE and analyzed by immunoblotting using anti-Ub (Enzo Life Sciences) or anti-myc antibodies (Sigma-Aldrich).

## MBP pull-down assays

MBP recombinant protein fusions were expressed in the *Escherichia coli* BL21 (DE3) strain carrying the corresponding coding sequence cloned into the pKM596 plasmid, a gift from David Waugh (Addgene plasmid # 8837). Recombinant proteins were purified and pull-down assays were performed according to (*Fonseca and Solano, 2013*). Exponentially-growing cultures (OD at 600 nm of 0.6) were induced with 0.2 mM isopropyl-D-thiogalactopyranoside for 4 hr, lysed using a sonicator, and centrifuged at 16,000 g for 30 min at 4°C. MBP-tagged fusions were purified using amylose agarose beads under native conditions as described by the manufacturer (Amersham). Equal amounts of seedling protein extracts were combined with 10 µg MBP-tagged fusion or MBP protein alone,

bound to amylose resin for 1 hr at 4°C with rotation, washed three times with 1 ml of extraction buffer, eluted and denatured in sample buffer before immunoblot analysis.

## TAP assays

Cloning of transgenes encoding GS[rhino] tag (*Van Leene et al., 2015*) fusions under control of the constitutive cauliflower tobacco mosaic virus 35S promoter and transformation of Arabidopsis cell suspension cultures with direct selection in liquid medium were carried out as previously described (*Van Leene et al., 2015*). TAP experiments were performed with 200 mg of total protein extract as input as described in *Van Leene et al. (2015)*. Protein interactors were identified by mass spectrometry using an LTQ Orbitrap Velos mass spectrometer. Proteins with at least two matched high confident peptides were retained. Background proteins were filtered out based on frequency of occurrence of the co-purified proteins in a large dataset compiling 543 TAP experiments using 115 different baits (*Van Leene et al., 2015*). Detailed protein identification obtained with the LTQ Orbitrap Velos (Thermo Fisher Scientific) or Q Exactive (Thermo Fisher Scientific) and Mascot Distiller software (version 2.5.0.0, Matrix Science) is given in *Figure 4—source data 1*.

## Histone proteomics analyses

Histone proteins samples were obtained from 50 mg of plant material by preparing chromatin-enriched fractions following the ChIP protocol (omitting the formaldehyde crosslinking step) and subsequent purification of 1 mg proteins re-suspended in ChIP Nucleus Lysis Buffer (SDS 1%, EDTA 10 mM, Tris pH8) using the Histone Purification Mini kit (Active Motif) following the manufacturer's instructions. Three independent histone purifications were simultaneously separated by SDS-PAGE and stained with colloidal blue (LabSafe Gel Blue GBiosciences). Three gel slices were excised for each purification and in-gel trypsin digested (Gold, Promega). Peptides extracted from each set were pooled and analyzed in triplicate by nanoLC-MS/MS using an Ultimate 3000 system (Dionex S. A.) coupled to an LTQ-Orbitrap XL mass spectrometer (Thermo Scientific). For data processing, LTQ-Orbitrap XL data were searched against the UniProtKB/Swiss-Prot *Arabidopsis thaliana* database using MascotTM through Proteome DiscovererTM (version 1.4, Thermo Scientific). Enzyme specificity was set to trypsin and a maximum of five miss-cleavages. Maximum allowed mass deviation was set to 2 ppm for monoisotopic precursor ions and 0.8 Da for MS/MS peaks. Variable modifications included M oxidation, N-term and K acetylation, C carbamidomethylation, K ubiquitination and methylation (mono, di and tri). All peptide/protein identification datasets were further processed using the Institut Curie-developed software myProMS version 3 (*Poullet et al., 2007*) in which the false discovery rate for peptide identification was set to <0.01. Label-free quantification and the identification of H2A and H2B monoubiquitination sites were as follows. To quantify the modified peptides, we extracted from the MS survey of nano-LC-MS/MS raw files the extracted ion chromatogram (XIC) signal of the well characterized tryptic peptide ions using the Pinpoint (version 1.2, Thermo Scientific) software. The peptide XIC areas were log2 transformed and the mean Log2 area was normalized by the mean area of non-modified peptide YNKKPTITSR for H2B using software R version 3.1.0. On each peptide a linear model was used to estimate the mean and the 5% confidence interval of each condition. The associated ratio and p-value have been computed thanks to a two-sided t-test with the Welch approximation of the number of freedom degree. Tryptic digest of ubiquitinated proteins degrades the ubiquitin modification similar to other proteins, resulting in an increment of 114 Da to the peptide mass by leaving a small di-Glycine signature peptide at the ubiquitination site. Excessive alkylation by iodoacetamide has been reported to potentially induce an artifact having the exact chemical composition ($C_4H_6N_2O_2$) as di-Gly. Consequently, iodoacetamide was substituted with chloroacetamide in order to avoid any in vitro artifacts when using mass spectrometry for diagnosis of the ubiquitination sites. MS/MS spectra shown in *Figure 1* unambiguously determined the H2A ubiquitination at Lys 129 and respectively at the Lys C-terminal for H2B. Detailed identification of differentially modified histone peptides in wild-type and *det1-1* samples is given in *Figure 1—source data 1*.

## Cytological analyses

Immunolocalization of fixed proteins in isolated nuclei was performed on 7-days-old seedlings as described in *Bourbousse et al., 2015*, using the antibodies given in Supplementary Procedures.

Images were acquired using a confocal laser-scanning microscope (SP5, Leica) and processed using ImageJ (rsb.info.nih.gov/ij/). For BiFC experiments, Agrobacterium strains expressing the indicated combinations of fusion proteins were co-infiltrated into the abaxial surface of 3-week-old *Nicotiana benthamiana* plants using the p19 protein to suppress gene silencing as described in *Irigoyen et al. (2014)*. Fluorescence was visualized in leaf epidermal cells 3 days after infiltration using a Leica SP5 confocal microscope.

## Size-exclusion chromatography

Ten-day-old seedlings were ground in liquid nitrogen and resuspended in 1 ml of extraction buffer (50 mM Tris–HCl pH 7.5, 150 mM NaCl, 5 mM EDTA, 0.1% NP-40, 10% glycerol and protease inhibitors). The extract was cleared by successive centrifugation for 5 min at 5,000 g, 5 min at 10,000 g and 1 hr at 16,000 g. The soluble fraction was subsequently concentrated at 14,000 rpm in YM-10 centricons (Amicon). About 250 µL (1 mg) of cleared extract was injected in a pre-calibrated Superdex 200 gel filtration column (GE Healthcare) and run with the same extraction buffer at 0.4 ml.min-1 in an AKTA FPLC system. Twenty 0.5 ml fractions were collected and analyzed by immunoblotting with an anti-GFP antibody (632381 Clontech).

## Yeast experiments

For yeast complementation assays, the *UBP22* coding sequence was amplified using primers AT-NOT1-22 and 22-PSTI-AT adding a Not1 and a Pst1 restriction site in 5' and 3', respectively, of the CDS and ligated into a pCRII-TOPO vector. UBP22 Not1/Pst1 restriction products were further inserted in the pCM185 plasmid (ATCC 87659). The yeast strain *ubp8Δ* (UCC6392; *Gardner et al., 2005*) was transformed by heat shock and total proteins from randomly spotted colonies were extracted using 8M urea SDS loading buffer to assess H2Bub levels by immunoblot. Primer sequences are given in Supplementary Procedures. Yeast-two-hybrids experiments were performed using the Matchmaker (Clontech) system according to the manufacturer instructions. Both bait and prey empty vectors were used as negative controls.

## Protein sequence analyses and modeling

The structure of UBP22, SGF11 and ENY2 were predicted using the Phyre2 web portal for protein modeling, prediction and analysis (http://www.sbg.bio.ic.ac.uk/phyre2/) (*Kelley et al., 2015*). The best models for AtUBP22 and AtSGF11 were based on the structure of ScUbp8 and ScSgf11 within the yeast DUBm at 1.89 Å resolution (PDB code 3MHS, *Samara et al., 2010*) with 100% confidence and 80% sequence coverage, and 99.9% confidence and 51% sequence coverage, respectively. AtENY2 best model was based on the structure of ScSus1 within the yeast TREX-2 complex at 2.7 Å resolution (PDB code 3FWC) with 99.9% confidence and 58% sequence coverage. In the later case, a structural homology search using DALI (*Holm and Laakso, 2016*) recognizes ScSus1 in the DUBm context with a high score (Z-score = 10.2 with a rmsd of 2.1 Å over 85 residues). Protein visualization was done using UCSF Chimera (https://www.cgl.ucsf.edu/chimera). A potential SGF73 plant homolog was searched by identifying proteins displaying partial or full-length similarities to *S. cerevisiae* SGF73 primary structure from the 33,090 non-redundant protein sequence database of green plants and the 3027 protein sequence database of algae (PSI-BLAST), or secondary structure from *Arabidopsis thaliana* (TAIR10) and *Chlamydomonas reihnardtii* (Jul_27_2017) proteomes (HHPRED; *Zimmermann et al., 2018*).

## ChIP-Rx analyses

ChIP experiments were conducted as described previously (*Bourbousse et al., 2012*) in two biological replicates of 5-day-old WT and *det1-1* seedling grown under light or constant dark conditions using an anti-H2Bub antibody (MM-0029-P, Medimabs). For each biological replicate, two IPs were carried out using 100 µg of Arabidopsis chromatin mixed with 3 µg of Drosophila chromatin, as quantified using BiCinchoninic Acid assay (Thermo Fisher Scientific). DNA eluted and purified from the two technical replicates was pooled before library preparation (Illumina TruSeq ChIP) and sequencing (Illumina sequencing single-reads, $1 \times 50$ bp) of the resulting 8 input and 8 IP samples by Fasteris (Geneva, Switzerland). Reads were mapped using STAR version 2.5.3a (STAR –alignIntronMax 1 –alignEndsType EndToEnd) (*Dobin et al., 2013*) onto combined TAIR10 *Arabidopsis*

*thaliana* and *Drosophila melanogaster* (dm6) genomes. Duplicate reads were removed. Consistency between biological replicates has been confirmed by Hierarchical dendrogram clustering based on Spearman correlation, by Heatmap representation based on Euclidean distance between samples, and by Principal component analysis (PCA). To calculate the Rx scaling factor of each biological replicate, Drosophila-derived IP read counts were normalized according to the number of Input reads. Spike-in normalization was performed as described in *Orlando et al. (2014)* with some modifications: $\alpha = r/Nd\_IP$, where Nd_IP corresponds to the number of reads (in millions) aligning to the *D. melanogaster* genome in the IP and r corresponds to the percentage of Drosophila-derived reads in the input. Differently from the original method that postulates an invariable amount of Drosophila chromatin in each input sample, we quantified the corresponding reads in each input to take into account technical variations between replicates. Hence, r was calculated for each replicate as $100*Nd\_i/(Na\_i + Nd\_i)$, where Nd_i and Na_i are the number of reads (in millions) aligning to the *D. melanogaster* or *A. thaliana* genome in the input, respectively. H2BUb enriched regions were obtained using MACS2 version 2.1.1.20160309 (callpeak -f BAM –keep-dup 1 g 1e + 8) (*Zhang et al., 2008*). For tracks visualization, IGV version 2.3.93 (*Thorvaldsdóttir et al., 2013*) browser tracks were obtained as described by (*Orlando et al., 2014*) by scaling MACS2 values in the bedgraph pilup files with the Rx normalization factors described above. Then, values obtained for replicates were averaged and converted to bigwig files. To determine H2BUb-marked genes, H2Bub enriched domains reproducibly detected with MACS2 in the two biological replicates with an FDR < 0.01 were intersected using Bedtools Utility Intersect version v2.26.0 (*Quinlan and Hall, 2010*). Only peaks exhibiting intersection regions beyond 10% of input peak lengths were kept. The resulting peaks from the 2 biological replicates were merged and annotated with TAIR10 gene coordinates (corresponding to −250 pb from gene TSS to TES). Only genes and TEs that had intersection regions greater than 150 bp with the resulting peaks were kept. To determine Differentially-Ubiquitinated Genes (DUGs), the number of reads mapping into the peak coordinates of each replicate samples was calculated using Bedtools Utility Multicov and the peaks from all 8 samples were grouped by gene-ID to obtain unique peak coordinates per marked gene using Bedtools Utility Groupby version v2.26.0 (Quinlan and Hall, 2010). The coverage values from all 8 samples were then scaled using Rx factors (*Figure 2—source data 1*), and used for DESeq2 version 1.19.37 (*Love et al., 2014*) analysis using Wald test. Source codes of ChIP differential analyses are given online as *Figure 2—source code 1*.

## RNA analyses

For RT-qPCR experiments, 1 μg of total RNA was isolated with the NucleoSpin RNA Plant kit (Macherey Nagel), treated with Amplification Grade DNaseI (Invitrogen), and cDNAs were synthesized using the High-Capacity cDNA Reverse Transcription kit (Applied Biosystems). Quantitative PCR was carried out using the LightCycler 480 SYBR Green I Master (Roche). *UBP22* transcripts were quantified using the primer pair UBP22_RT1_Fwd and UBP22_RT1_Rev and results were normalized to *ACT2* transcript levels using ACTIN2_CF_F and ACTIN2_CF_R primers. For RNA-seq experiments, total RNAs from 5-day-old seedlings independently grown in two biological replicates were extracted using the Quick-RNA MiniPrep kit (Zymo) and processed independently for library preparation and sequencing (Illumina sequencing single-reads, 1 × 50 bp) by Fasteris (Geneva, Switzerland) using 1 μg of RNA. Single-end RNA-Seq reads were mapped onto the TAIR10 *Arabidopsis thaliana* genome using STAR (*Dobin et al., 2013*) version 2.5.3a (STAR –outFilterMismatchNmax 2 –outFilterMultimapNmax 1000 –alignIntronMin 20 –alignIntronMax 6000). Expression levels of individual genes were obtained using HTSeq (*Anders et al., 2015*) version 0.7.2 (htseq-count -f bam) and Reads Per Kilobase Million (RPKM) were calculated for each gene. Consistency between biological replicates has been confirmed by Hierarchical dendrogram clustering based on Spearman correlation, by Heatmap representation based on Euclidean distance between samples, and by Principal component analysis (PCA). Differential expression analysis was performed using DESeq2 version 1.19.37 (*Love et al., 2014*) on raw read counts to obtain normalized fold changes (FC) and *Padj*-values for each gene. Genes were considered to be differentially expressed only if they showed a log2FC >1 and a FDR < 0.01 in Wald test. Source codes of RNA-seq differential analyses are given online as *Figure 2—source code 2*.

## GEO accession

ChIP-seq and RNA-seq data generated in this work are accessible through GEO Series accession number GSE112952.

## Antibodies

Antibodies used for immunoblot experiments: Anti-GFP-HRP (Milteny Biotec #130-091-833); monoclonal anti-FLAG M2 (Sigma-Aldrich #F3165); anti-HA-HRP (Roche #3F10), anti-H2B (Millipore #07–371) or kindly provided by Prof. Spiker, anti-H2Bub (Medimabs #MM-0029); anti-H3K4me2 (Millipore #07–030); anti-H3K4me3 (Millipore #05–745); anti-H3K9me1 (Millipore #07–450); anti-H3K27me3 (Millipore #07–449); anti-H3K36me3 (Millipore #07–353); anti-H3ac (Millipore #06–599); anti-H3K9ac (Millipore #06–942); anti-H3K27ac (Millipore #07–360); anti-H4ac (Millipore #06–598); anti-H3 (Millipore #05–499); anti-RPT5 (Enzo Life Sciences# BML-PW8245). Antibodies used in cytological analyses: Anti-MYC (Millipore #05–724) or anti-GFP (ThermoFisher Scientific #A11122) primary antibodies, Alexa-488 coupled anti-mouse (ThermoFisher Scientific #A11001) or anti-rabbit (ThermoFisher Scientific #A11008) secondary antibodies. Antibodies used in ChIP-seq analyses: anti-H2Bub (Medimabs #MM-0029-P).

## Oligonucleotide sequences

UBP22gabiF 5'-GCAGCTTTTTGGTTTGGGTA-3'; UBP22gabiR 5'-TGTTGACCACTCAACCAAAAG-3'; GABI o8409 5'-ATATTGACCATCATACTCATTGC-3'; UBP22_RT1_Fwd 5'-GTCCGCGAGGATTTCATTT-3'; UBP22_RT1_Rev 5'-TTGCATCGTTGAAGCATCTG-3'; AT-NOT1-22 5'-ATGCGGCCGCATGTCCGCGAGGATTTCATTTC-3'; 22-PSTI-AT 5'-TACTGCAGTCAGCAATCAGCAAAGGAAAT-3'; ACTIN2_CF_F 5'-TCAGATGCCCAGAAGTGTTGTTCC-3'; ACTIN2_CF_R 5'-CCGTACAGATCCTTCCTGATATCC-3'; UBP22_FCl CACCATGTCCGCGAGGATTTC; UBP22_RCl TCAGCAATCAGCAAAGGGAAATGC; SGF11.BF 5'-GGGGACAAGTTTGTACAAAAAAGCAGGCTATATGTCTGGCGCAGAGGATAATAAAT-3'; SGF11.BR 5'-GGGGACCACTTTGTACAAGAAAGCTGGGTATCAGTCTCCTTTCACGTTCTCTCG-3'; ENY2.BF 5'-GGGGACAAGTTTGTACAAAAAAGCAGGCTATATGAAACATTCGGTGAATCG-3'; ENY2.BR 5'-GGGGACCACTTTGTACAAGAAAGCTGGGTATCAAAACACCATCAAAGAGC-3'.

# Acknowledgements

The authors gratefully acknowledge Alexandre Sta (Institut Curie, Paris, France) and Magali Charvin (IBENS, Paris, France) for technical assistance, Jérome Deraze and Frédérique Perronet (Institut de Biologie Paris Seine, France) for providing Drosophila samples, Daniel E Gottschling (Fred Hutchinson Cancer Research Center, USA) for providing the ubp8Δ yeast strain (*Gardner et al., 2005*) and Lionel Schiavolin for unpublished work (IBENS, Paris, France). Authors are much indebted to Roberto Solano (CNB, Madrid, France), Crisanto Gutiérrez (CBM Severo Ochoa, Madrid, Spain), Giovanna Benvenuto (Stazione Zoologica, Naples, Italy), Daniel Bouyer and Vincent Colot (IBENS, Paris, France) for helpful discussions. They also thank the European Union COST Actions CA16212 INDEPTH and BM1307 PROTEOSTASIS.

# Additional information

## Funding

| Funder | Grant reference number | Author |
| --- | --- | --- |
| Agence Nationale de la Recherche | ANR-11-JSV2-003-01 | Fredy Barneche |
| Agence Nationale de la Recherche | ANR-10-LABX-54 | Chris Bowler |
| Agence Nationale de la Recherche | ANR-10-IDEX-0001-02 | Chris Bowler |
| Université Paris-Sud | PhD Fellowship | Martin Rougée |
| Université Paris-Sud | PhD Fellowship | Clara Bourbousse |

| | | |
|---|---|---|
| Ministerio de Economía y Competitividad | Grant BIO2013-46539-R | Vicente Rubio |
| Agencia Estatal de Investigación/Fondo Europeo de Desarollo regional/European Union | Grant BIO2016-80551-R | Vicente Rubio |
| Fundación Bancaria Caixa d'Estalvis i Pensions de Barcelona | PhD Fellowship | Amr Nassrallah |
| Ministerio de Economía y Competitividad | PhD Fellowship | Elisa Iniesto |
| Ministerio de Economía y Competitividad | Ramón y Cajal grant RYC-2014-16308 | Sandra Fonseca |

The funders had no role in study design, data collection and interpretation, or the decision to submit the work for publication.

## Author contributions

Amr Nassrallah, Stephanie Drevensek, Elisa Iniesto, Berangere Lombard, Resources, Data curation, Formal analysis, Validation, Investigation, Visualization, Methodology; Martin Rougée, Clara Bourbousse, Conceptualization, Resources, Data curation, Formal analysis, Validation, Investigation, Visualization, Methodology; Sandra Fonseca, Conceptualization, Resources, Data curation, Formal analysis, Funding acquisition, Validation, Investigation, Visualization, Methodology; Ouardia Ait-Mohamed, Ikhlak Ahmed, Vanessa Masson, Resources, Data curation, Software, Formal analysis, Validation, Investigation, Visualization, Methodology; Anne-Flore Deton-Cabanillas, Gerald Zabulon, Resources, Data curation, Investigation, Visualization, Methodology; David Stroebel, Resources, Data curation, Investigation, Methodology; Dominique Eeckhout, Resources, Data curation, Formal analysis, Validation, Investigation, Visualization, Methodology, Writing—review and editing; Kris Gevaert, Investigation, Methodology; Damarys Loew, Conceptualization, Resources, Data curation, Formal analysis, Supervision, Validation, Investigation, Methodology, Writing—original draft, Writing—review and editing; Auguste Genovesio, Supervision, Investigation, Visualization, Methodology; Cecile Breyton, Conceptualization, Formal analysis, Validation, Investigation, Visualization, Methodology, Writing—original draft, Writing—review and editing, Performed protein sequence modeling; Geert De Jaeger, Conceptualization, Resources, Data curation, Supervision, Validation, Investigation, Visualization, Methodology, Writing—review and editing; Chris Bowler, Conceptualization, Supervision, Funding acquisition, Investigation, Visualization, Project administration, Writing—review and editing; Vicente Rubio, Conceptualization, Resources, Data curation, Formal analysis, Supervision, Funding acquisition, Validation, Investigation, Visualization, Methodology, Writing—original draft, Project administration, Writing—review and editing; Fredy Barneche, Conceptualization, Resources, Formal analysis, Supervision, Funding acquisition, Validation, Investigation, Visualization, Methodology, Writing—original draft, Project administration, Writing—review and editing, FB, ChB and VR conceived the experiments in close interaction with all their laboratory members, rose funding and wrote the manuscript

## Author ORCIDs

Amr Nassrallah (iD) https://orcid.org/0000-0002-2229-2855
Sandra Fonseca (iD) http://orcid.org/0000-0002-2021-4482
Elisa Iniesto (iD) http://orcid.org/0000-0003-1662-7361
Chris Bowler (iD) https://orcid.org/0000-0003-3835-6187
Vicente Rubio (iD) http://orcid.org/0000-0002-8800-2400
Fredy Barneche (iD) http://orcid.org/0000-0002-7014-7097

## Decision letter and Author response

Decision letter https://doi.org/10.7554/eLife.37892.030
Author response https://doi.org/10.7554/eLife.37892.031

## Additional files

### Data availability

As indicated in the manuscript, sequencing data have been deposited in GEO under accession codes GSE112951 (ChIP-seq) and GSE112950 (RNA-seq). All datasets from this study are combined in a super-series (GSE112952). All other data generated or analyzed during this study are included in the manuscript and supporting files. Source data files have been provided for Figures 1, 2, 4 and 5. The most relevant bioinformatics source code files for Figures 2 and 7 have been provided as individual files.

The following datasets were generated:

| Author(s) | Year | Dataset title | Dataset URL | Database and Identifier |
|---|---|---|---|---|
| Clara Bourbousse, Ouardia Ait-Mohamed | 2018 | Nassrallah, Rougée et al., ChIP-seq datasets | https://www.ncbi.nlm.nih.gov/geo/query/acc.cgi?acc=GSE112951 | NCBI Gene Expression Omnibus, GSE112951 |
| Clara Bourbousse, Ouardia Ait-Mohamed, Martin Rougée, Fredy Barneche | 2018 | Nassrallah, Rougée et al., ChIP-seq and RNA-seq super-series | https://www.ncbi.nlm.nih.gov/geo/query/acc.cgi?acc=GSE112952 | NCBI Gene Expression Omnibus, GSE112952 |
| Bourbousse C, Ait-Mohamed O, Rougée M, Barneche F | 2018 | Nassrallah, Rougée et al., RNA-seq datasets | https://www.ncbi.nlm.nih.gov/geo/query/acc.cgi?acc=GSE112950 | NCBI Gene Expression Omnibus, GSE112950 |

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
