## [Decision Letter]

Thank you for submitting your article "DET1-mediated degradation of a SAGA-like deubiquitination module controls H2Bub homeostasis" for consideration by *eLife*. Your article has been reviewed by two peer reviewers, and the evaluation has been overseen by Reviewing Editor Richard Amasino and Senior Editor Christian Hardtke. The following individuals involved in review of your submission have agreed to reveal their identity: Eirini Kaiserli and Klaus Grasser.

The Reviewing Editor has drafted the decision based on discussions of your article to help you prepare a revised submission. The only experiments we ask involve some simple de-etiolation assays.

Your paper reports interesting and novel results that provide evidence for a direct link between DET1 and histone H2B mono-ubiquitination in *Arabidopsis* and is suitable for publication in *eLife*.

There is one experiment that we think is relatively straightforward that should be done prior to publication: to include some basic de-etiolation assays (hypocotyl elongation upon white light irradiation) on the *ubp22-1, sgf11-1*, and *sgf11-2* single mutants, the *det1-1 ubp22-1* double mutant, and the UBP22 and SGF11 over-expressing lines described in the paper.

Another key point prior to publication is that the discussion (subsection “An atypical H2Bub DUB module is targeted by light signaling for ubiquitin-mediated proteolysis”) of the unusual *Arabidopsis* DUBm (compared to that of yeast/metazoa) would be strengthened by discussing the results more thoroughly in the context of the related findings reported recently by Pfab et al., 2018 in which the composition of the DUBm and its H2B de-ubiquitination activity were described. Both reports point out consistent differences between plant (*Arabidopsis*) SAGA/DUBm and the wealth of literature on yeast/metazoan SAGA. Discussing Pfab et al., 2018 in your paper will underscore the novel findings regarding plant SAGA/DUBm and provide an opportunity for synthesis for readers of your paper.

Finally, the *sgf11-2* allele reported in your paper is termed *sgf11-1* in the article by Pfab et al. (and in TAIR). To avoid confusion and to follow precedent, you should rename the two alleles in this manuscript accordingly by changing your *sgf11-2* to *sgf11-1*.

---

## [Author Response]

There is one experiment that we think is relatively straightforward that should be done prior to publication: to include some basic de-etiolation assays (hypocotyl elongation upon white light irradiation) on the ubp22-1, sgf11-1, and sgf11-2 single mutants, the det1-1 ubp22-1 double mutant, and the UBP22 and SGF11 over-expressing lines described in the paper.

In response to the request, the revised manuscript provides the results of de-etiolation assays in which typical measurements of seedling hypocotyl length upon growth in darkness were performed. Although global H2Bub levels cannot directly be linked to photomorphogenic phenotypes (see our previous work in Bourbousse et al., 2012), these novel analyses are in good agreement with data obtained at the molecular level for the *det1-1* and *det1-1ubp22-1* mutant lines. Especially, the *ubp22-1* mutation partially suppressed the constitutive photomorphogenic phenotype (short hypocotyl in darkness) of *det1-1* seedlings, whereas the *ubp22-1* mutation or UBP22 overexpression on their own do not affect hypocotyl length. Hence, skotomorphogenic phenotype of *det1-1ubp22-1* double mutant plants nicely relates to the partial suppression of *det1-1* global transcriptomic defects by the *ubp22-1* mutation observed under light conditions (Figure 7A and 7B). These results have been included in a new panel of Figure 7.

As requested, we also analyzed *sgf11-2* and SGF11 overexpression lines. Again, these analyses did not allow identifying significant differences as compared to wild-type controls when SGF11 content is modulated in a wild-type background. Considering that this latter information does not help in the current understanding of the DUBm function, the corresponding data has not included in the manuscript but could be added upon request.

We also appreciate that more detailed characterization of the photomorphogenic responses linked to histone H2B ubiquitination/deubiquitination is of interest. This is currently ongoing in our laboratories, aiming at determining the effect of different light signaling pathways in C3D function towards the DUBm. Considering the complexity and duration of our investigations, we believe that further investigations in that direction fall beyond the scope of this manuscript.

Another key point prior to publication is that the discussion (subsection “An atypical H2Bub DUB module is targeted by light signaling for ubiquitin-mediated proteolysis”) of the unusual Arabidopsis DUBm (compared to that of yeast/metazoa) would be strengthened by discussing the results more thoroughly in the context of the related findings reported recently by Pfab et al., 2018 in which the composition of the DUBm and its H2B de-ubiquitination activity were described. Both reports point out consistent differences between plant (Arabidopsis) SAGA/DUBm and the wealth of literature on yeast/metazoan SAGA. Discussing Pfab et al., 2018 in your paper will underscore the novel findings regarding plant SAGA/DUBm and provide an opportunity for synthesis for readers of your paper.

We also concur with this request. As informed during the initial submission, the manuscript by Pfab et al. (2018) describing the *Arabidopsis* DUBm organization has been released online while we were also submitting our manuscript. This is why Pfab et al. was shortly cited in the Discussion. In this revised manuscript, we have included much more detailed references to similar or complementary findings by Pfab et al. (2018) at several places in the Results and Discussion sections.

Finally, the sgf11-2 allele reported in your paper is termed sgf11-1 in the article by Pfab et al. (and in TAIR). To avoid confusion and to follow precedent, you should rename the two alleles in this manuscript accordingly by changing your sgf11-2 to sgf11-1.

Yes, this is due to concomitant work on those lines in each laboratory. Any confusion has now been adjusted in all instances, such as in the Results section, in Figure 5 and Figure 5—figure supplement 1 as well as in the Materials and methods section by citing Pfab et al. (2018) for the SAIL_856_F11 mutant line.